# ORION: INTERVENTION-VALIDATED CAUSAL DISCOVERY FOR SPATIOTEMPORAL FORECASTING

## ABSTRACT

While spatiotemporal forecasting plays a critical role in domains such as intelligent transportation and infectious disease prediction, existing methods still face fundamental challenges in modeling complex dependencies in graph-structured data and multi-scale temporal dynamics, primarily because they neglect the inherent causal mechanisms in spatiotemporal propagation processes. To address these limitations, we propose Orion, a framework that integrates trainable causal inference with attention mechanisms, optimized through progressive training. Our key contributions include: (1) the TE-CausGAT module, which jointly performs causal discovery and validates learned structures via do-interventions; (2) the Belt Block architecture, which processes hourly, daily, and weekly patterns in parallel; (3) intelligent data retrieval using a fine-tuned large language model; and (4) a progressive three-stage training strategy that progressively transitions from feature learning to causal discovery to end-to-end optimization. Orion achieves new state-of-the-art performance across multiple benchmarks: in traffic forecasting, it improves MAE by 6.60% on METR-LA, 8.65% on PEMS-08, and shows strong short-horizon gains and competitive long-horizon results on PEMS-04; in epidemic forecasting, it improves MAE by 22.70% on CA and 37.73% on TX. Furthermore, comprehensive ablation studies, Dream3 causal validation experiments, and METR-LA case studies collectively demonstrate Orion's effectiveness in both accurate prediction and reliable causal discovery.

## 1 INTRODUCTION

Spatiotemporal forecasting is the cornerstone for understanding and modeling the evolution of complex dynamical systems, playing a pivotal role in numerous domains including intelligent transportation systems (ITS) and infectious disease prediction. The underlying mechanisms driving these systems extend far beyond simple statistical correlations. Real-world spatiotemporal systems universally exhibit an often-overlooked characteristic: their evolution is governed by latent causal mechanisms that determine how perturbations propagate, amplify, or dissipate throughout the network, directly impacting both prediction accuracy and interpretability.

To concretely illustrate this challenge, we examine a representative scenario of highway network dynamics during the morning rush hour in an urban interchange area, as depicted in Figure 1. At 7:30 AM, an accident occurs at Node A, causing traffic flow to plummet from normal levels to merely 30 vehicles per 5-minute interval, thereby establishing the initial congestion source. This perturbation triggers non-uniform spatiotemporal propagation. First, the causal impact exhibits distance-dependent temporal delays: upstream Node B experiences a flow reduction to 80 vehicles within 5 minutes, while distant Node C requires 8 minutes to show significant deceleration. Second, the system demonstrates asymmetric upstream-downstream effects: downstream Node D paradoxically sees an increase in flow to 220 vehicles due to the "downstream release" phenomenon. Third, the network topology creates indirect causal chains: by 7:50 AM, the congestion propagates through ramp connections, affecting previously uncorrelated nodes, with Node E's flow dropping to 110 vehicles and Node F's flow decreasing to 125 vehicles due to traffic redistribution. These spatiotemporal causal characteristics also serve as the foundation for subsequent modeling.

Existing spatiotemporal forecasting methods have evolved from generic architectures to specialized designs. Early studies explored applying general deep learning architectures to spatiotemporal data: RNNs(Schuster & Paliwal, 1997) model temporal dependencies through sequence modeling, Transformers (Vaswani et al., 2017) capture long-range temporal dependencies through global attention mechanisms; convolution-based GCN (Li et al., 2018) and TCN (Lea et al., 2016) capture local patterns in graph-structured data through spatial convolutions, and attention-based graph networks such as GAT (Veličković et al., 2017) learn graph structural representations via message passing. However, these general architectures lack spatiotemporal coupling capabilities and struggle to capture complex spatial–temporal interactions simultaneously. The second category of methods designs coupled architectures specifically for spatiotemporal forecasting: DCRNN (Li et al., 2017) integrates diffusion convolution and GRU for spatiotemporal modeling, GWNet (Wu et al., 2019) introduces adaptive adjacency matrix learning, and AGCRN (Bai et al., 2020) and DGCRN (Li et al., 2023) achieve spatiotemporal coupling through message passing. However, these methods typically assume static graph structures that cannot adapt to dynamic evolution and treat inter-node dependencies as bidirectional symmetric relationships, ignoring causal directionality. The third category of methods begins to integrate causal inference: CaST (Xia et al., 2023) improves forecasting through causal graph learning, and TESTAM (Lee & Ko, 2024) performs causal discovery using spatiotemporal attention. However, these methods rely on fixed causal graphs and strong assumptions, making it difficult to truly leverage and validate the discovered causal relationships.The complete "Related Work" section can be found in Appendix A.

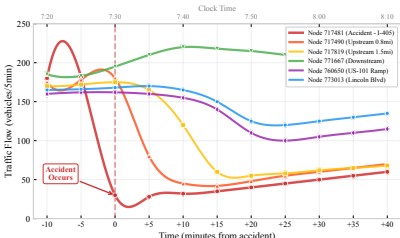

(a) Temporal distribution shifts

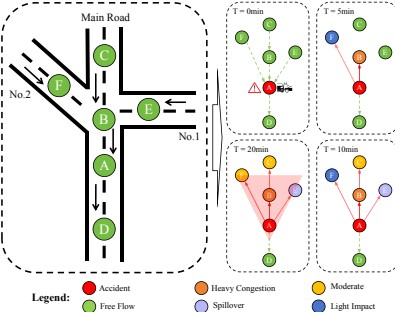

(b) Spatial causal relationship propagation

Figure 1: Traffic flow pattern analysis.

Despite advances in specific scenarios, these methods still exhibit four categories of limitations: (1) neglecting causal directionality, unable to distinguish between causal sources and targets; (2) static dependency assumptions hindering dynamic structural adaptation or ignoring inherent spatial patterns; (3) single-scale temporal modeling struggling to capture multi-scale features; (4) causal models still rely on strong assumptions and fixed graphs without reliable validation. We propose Orion to address these issues, with the following key contributions:

- We propose an LLM-enhanced retrieval mechanism, fine-tuned on TinyLlama-1.1B-Chat-v1.0, that selects historical periods by semantic similarity.

- We design the TE-CausGAT module that performs temporal causal estimation, adjacency fusion, and do-intervention validation, enabling autonomous discovery and validity verification of causal relationships.

- We introduce the Belt Block combining TE-CausGAT and multi-head attention to parallelly integrate multi-scale temporal information.

- We develop a three-stage training paradigm based on curriculum learning principles that smoothly transitions from feature learning to causal discovery and joint optimization.

- Orion achieved SOTA results across five datasets, with additional experiments collectively demonstrating its exceptional performance, generalization capabilities, and effective causal discovery.

## 2 MOTIVATION FOR DYNAMIC CAUSAL REASONING

Before introducing the specific modules of Orion, it is necessary to explain why spatiotemporal forecasting urgently needs to shift from correlation-based learning to causal reasoning. Although existing methods can capture statistical patterns, they suffer from fundamental limitations when facing distribution shifts, interpretability requirements, and dynamic changes. Causal reasoning can

extract the intrinsic mechanisms of systems and provide a more robust modeling framework. The necessity of this paradigm shift is reflected in three aspects:

**Cross-distribution generalization**, causal relationships reflect the intrinsic generative mechanisms of a system and remain stable across distributions, whereas correlation patterns often fail outside the training distribution. This explains why traditional models perform well in routine scenarios but degrade sharply during abnormal events (e.g., sudden congestion, extreme weather).

**Interventability:** causal models not only predict "what will happen," but also reveal "why it happens" and "how to intervene." Traffic managers need to know which intersection to regulate to effectively alleviate regional congestion, and epidemic control requires identifying which measures can block transmission chains—these are fundamentally causal questions that correlation analysis cannot answer.

**Modeling temporal variability:** causal mechanisms in real systems are not fixed but dynamically evolve with time and context. During peak hours, a certain intersection may become the causal source of congestion and influence a wide region, while at night its causal impact range shrinks significantly. Ignoring such temporal variability leads to inconsistent model performance across different periods.

## 3 PRELIMINARIES

**Definition 1 (Spatiotemporal Forecasting Problem).** A spatiotemporal network is defined as $\mathcal{G} = (\mathcal{V}, \mathcal{E}, \mathbf{A})$, where $\mathcal{V} = \{v_1, v_2, ..., v_N\}$ denotes the set of $N$ spatial nodes, $\mathcal{E} \subseteq \mathcal{V} \times \mathcal{V}$ represents inter-node connections, and $\mathbf{A} \in \mathbb{R}^{N \times N}$ is the adjacency matrix. At time $t$, each node $v_i$'s state is represented by feature vector $\mathbf{x}_i^{(t)} \in \mathbb{R}^F$, with the entire network state denoted as $\mathbf{X}^{(t)} = [\mathbf{x}_1^{(t)}, \mathbf{x}_2^{(t)}, ..., \mathbf{x}_N^{(t)}]^T \in \mathbb{R}^{N \times F}$. Given historical observations over $T$ time steps $\mathcal{X} = \{\mathbf{X}^{(t-T+1)}, \mathbf{X}^{(t-T+2)}, ..., \mathbf{X}^{(t)}\}$, the spatiotemporal forecasting task aims to learn a mapping function $f : \mathbb{R}^{N \times F \times T} \times \mathcal{G} \to \mathbb{R}^{N \times H}$ that predicts future states over $H$ horizons $\hat{\mathcal{Y}} = \{\hat{\mathbf{X}}^{(t+1)}, \hat{\mathbf{X}}^{(t+2)}, ..., \hat{\mathbf{X}}^{(t+H)}\}$, minimizing prediction error $\mathcal{L}(\hat{\mathcal{Y}}, \mathcal{Y})$.

**Definition 2 (Dynamic Causal Graph).** A dynamic causal graph is defined as a sequence of time-varying directed acyclic graphs $\mathcal{C} = \{\mathbf{C}^{(s)}\}_{s=1}^S$, where $S$ denotes the number of temporal segments and $\mathbf{C}^{(s)} \in [0, 1]^{N \times N}$ represents the causal adjacency matrix for segment $s$. Matrix element $\mathbf{C}_{ij}^{(s)}$ quantifies the causal strength from node $v_i$ to node $v_j$ during segment $s$. Causal matrices exhibit directionality ($\mathbf{C}_{ij}^{(s)} \neq \mathbf{C}_{ji}^{(s)}$) and enforce the acyclicity constraint via polynomial approximation ($\text{tr}(e^{\mathbf{C}^{(s)}}) - N = 0$) (Zheng et al., 2018).

**Definition 3 (Causal Intervention and do-operator).** Intervention on node $v_i$ is defined as $do(X_i = x)$, which sets node $i$'s value to $x$ while severing all incoming causal edges. The post-intervention distribution $P(Y|do(X_i = x))$ differs from the observational conditional distribution $P(Y|X_i = x)$. Under causal graph $\mathbf{C}$, intervention effects are computed through truncated causal paths: $P(X_j|do(X_i = x)) = \sum_{X_{pa(j)}} P(X_j|X_{pa(j)}) \prod_{k \in pa(j) \setminus \{i\}} P(X_k)$, where $pa(j)$ denotes the parent set of node $j$.

## 4 METHODOLOGY

As shown in Figure 2, during Orion's forward propagation, the input data generates three period tensors $\mathbf{X}_h, \mathbf{X}_d, \mathbf{X}_w \in \mathbb{R}^{N \times F \times L}$ through the LLM-enhanced retrieval mechanism, which are then transformed into embedded representations $\mathbf{E} \in \mathbb{R}^{N \times d_{model} \times L}$ through embedding layers with positional encodings added. Three parallel Belt Blocks independently process period data, sequentially performing TE-CausGAT spatial attention, multi-head temporal attention, and feedforward network operations, interspersed with residual connections (He et al., 2016) and layer normalization (Ba et al., 2016). The fusion module receives branch outputs $\{\mathbf{H}_h, \mathbf{H}_d, \mathbf{H}_w\} \in \mathbb{R}^{N \times d_{model}}$, integrates three-period features through adaptive weighting and attention mechanisms, and generates predictions $\mathbf{X}_{output} \in \mathbb{R}^{N \times H}$ through linear projection. Notably, Orion adopts three-stage training: the first stage learns feature representations, the second stage introduces causal regularization with frozen embedding layers, and the third stage unfreezes parameters for end-to-end fine-tuning.

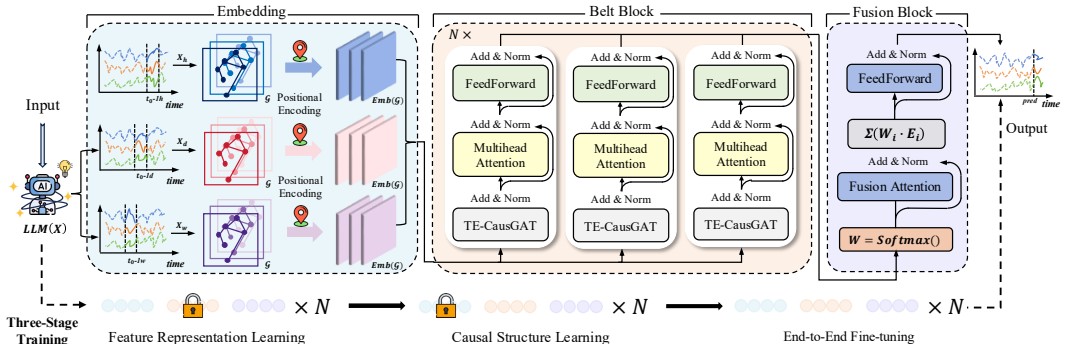

Figure 2: Overall architecture of the Orion framework.

## 4.1 Fine-tuned LLM-enhanced Data Retrieval

Traditional methods rely on fixed temporal windows to select historical reference data, which performs poorly when encountering non-periodic variations. For example, traffic patterns during holidays differ significantly from those on regular weekdays, and fixed retrieval of "the same time last week" may introduce irrelevant pattern noise. To overcome this limitation, we propose a semantic similarity-based intelligent retrieval mechanism. By fine-tuning the TinyLlama-1.1B model on 100,000 temporal samples from the PEMS03 dataset (details in Ap-

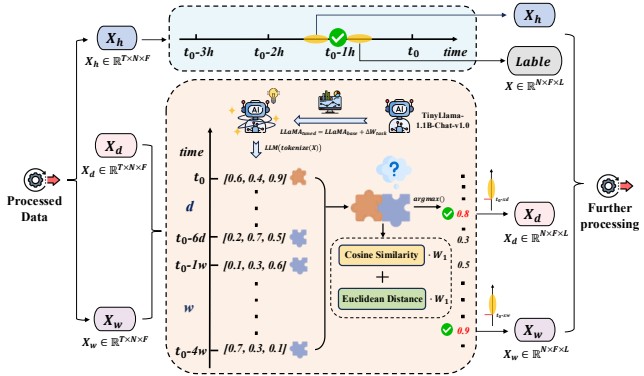

Figure 3: fine-tuned LLM-enhanced data retrieval.

pendix D), the LLM learns to understand semantic characteristics of temporal patterns, thereby enabling retrieval of truly similar historical patterns across calendar boundaries. As shown in Figure 3, taking the daily cycle as an example, both the input sequence $\mathbf{X}_h \in \mathbb{R}^{N \times F \times T}$ at current time $t$ and the data from the same time periods over the past $D$ days are encoded using a fine-tuned large language model, extracting the average of the last hidden layer as the semantic embedding $\mathbf{z}_i \in \mathbb{R}^{d_{embed}}$. The similarity between the current sequence and candidate historical sequences is measured through a weighted combination of cosine similarity and Euclidean distance:

$$S_{sim}(i,j) = w_{cos} \cdot \frac{\mathbf{z}_i^T \mathbf{z}_j}{||\mathbf{z}_i||_2 \cdot ||\mathbf{z}_j||_2} + w_{euc} \cdot \frac{1}{1 + ||\mathbf{z}_i - \mathbf{z}_j||_2} \tag{1}$$

Each node selects its most similar historical cycle as $\mathbf{X}_d$. This method captures overall patterns, handles irregular variations, and generalizes across data types. It is important to emphasize that, with or without the LLM, all input data are embedded through the same fully connected layer (nn.Linear) with parameter dimensions $F \times d_{\text{model}}$, followed by positional encoding before being processed by the Belt Block. The role of the LLM is intelligent data selection rather than feature extraction.

## 4.2 Temporal Evolution Causal Graph Attention Network (TE-CausGAT)

TE-CausGAT is core for causal discovery and validation in Orion. As in Figure 4, the pipeline proceeds from multi-scale temporal convolution to segmentation, causal estimation, Bayesian fusion, intervention validation, ending with causal-modulated attention. We denote $\mathbf{C}^{(s)}$, $\mathbf{C}_{\text{fused}}$, and $\mathbf{C}_{\text{prior}}$ as segment, fused, and prior causal matrices, bolded to distinguish from scalars; these represent prediction-oriented structures specific to each temporal segment rather than invariant causal relationships.

**Spatiotemporal Causal Discovery and Propagation.** TE-CausGAT applies eight-kernel convolutions on $\mathbf{X}_{input}$, generating multi-scale features via fusion. Next, Temporal segmentation divides the series into $S$ segments to capture causal evolution. Each segment $s$ uses an independent estimator for its causal structure. For segment $s$, causal relationship strengths are computed by learning source and target node transformation matrices through time-weighted feature aggregation combined with node embeddings:

$$\mathbf{C}^{(s)} = \sigma(\mathbf{S}^{(s)} \cdot (\mathbf{T}^{(s)})^T + \mathbf{B}) \quad (2)$$

Where $\mathbf{S}^{(s)}, \mathbf{T}^{(s)}$ are source and target transforms, $\mathbf{B}$ is a bias matrix, $\sigma$ is the sigmoid function, and the self-loop mask removes diagonal elements. The Bayesian causal fusion mechanism integrates segment-

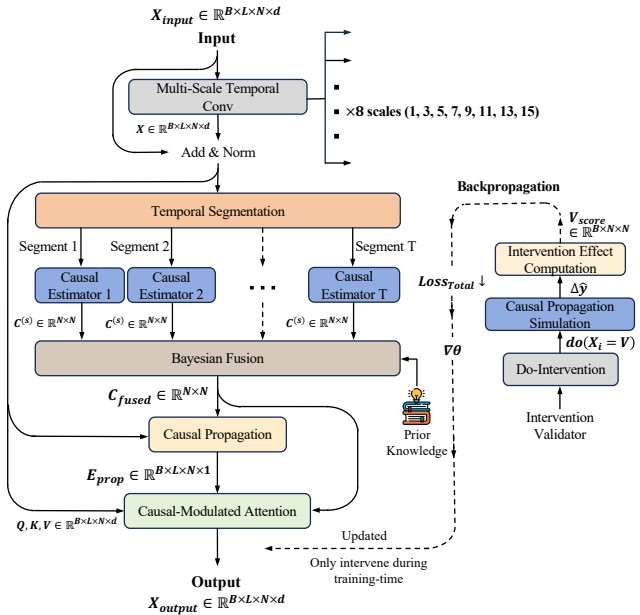

Figure 4: Architecture of the TE-CausGAT.

specific causal matrices through attention weights (the prior graph structure does not explicitly contain causal relationships) and incorporates prior structures from the adjacency matrix:

$$\mathbf{C}_{\text{fused}} = (1 - \alpha_{\text{prior}}) \cdot \sum_{s=1}^{S} w_s \cdot \mathbf{C}^{(s)} + \alpha_{\text{prior}} \cdot \mathbf{C}_{\text{prior}} \quad (3)$$

where $w_s$ denotes the attention weight for segment $s$ satisfying $\sum_{s=1}^{S} w_s = 1$, $\mathbf{C}_{\text{prior}}$ the prior adjacency, and $\alpha_{\text{prior}}$ balances discovered vs prior structures for stability. Subsequently, the causal propagation simulator employs a GRU mechanism (Cho et al., 2014) to model the temporally dynamic propagation of do-interventions:

$$\mathbf{h}^{(t)} = \text{GRU}(\mathbf{C}_{\text{fused}} \cdot \mathbf{X}^{(t)}, \mathbf{h}^{(t-1)}) \quad (4)$$

where $\mathbf{X}^{(t)} \in \mathbb{R}^{N \times d}$ is the node features at time $t$, $\mathbf{h}^{(t)} \in \mathbb{R}^{N \times d}$ is the hidden state, and the matrix product $\mathbf{C}_{\text{fused}} \cdot \mathbf{X}^{(t)}$ modulates information flow between nodes. The causal matrix modulates information flow between nodes and simulates the temporal propagation effects of node changes. To ensure the learned causal graph maintains acyclicity, we employ computationally efficient polynomial approximations instead of the theoretically exact but computationally expensive matrix exponential constraint $\text{tr}(e^{\mathbf{C}^{(s)}}) - N = 0$. Specifically, we penalize the trace of powers of the normalized causal matrix: $\mathcal{L}_{DAG} = \lambda_2 \cdot \text{tr}(\hat{\mathbf{C}}^2) + \lambda_3 \cdot \text{tr}(\hat{\mathbf{C}}^3)$, where $\hat{\mathbf{C}} = \mathbf{C}_{\text{fused}} / \|\mathbf{C}_{\text{fused}}\|_\infty$, $\lambda_2 = 0.1$ and $\lambda_3 = 0.01$. This exploits $\text{tr}(\mathbf{C}^k)$ counting $k$-cycles, suppressing cycles while remaining differentiable and efficient. Although the polynomial approximation cannot theoretically guarantee complete acyclicity, the causal graphs discovered by Orion strictly satisfy the DAG property in subsequent experimental learning (see in Appendix B.5 for details).

**do-intervention Validation.** The core feature of TE-CausGAT is a theoretically grounded do-intervention validation. Figure 5 illustrates how our do-intervention mechanism tracks causal effect propagation through the spatiotemporal network. When intervening on node $i$, the effect on downstream node $j$ can be expressed under linear approximation as $\Delta \hat{y}_j \approx \mathbf{C}_{ij} \cdot \Delta \mathbf{X}_i + \sum_{k \neq i} \mathbf{C}_{kj} \cdot \Delta \mathbf{X}_k$. We implement this mechanism through four steps: randomly selecting $K$ intervention nodes for unbiased discovery; generating intervention values $v = \tanh(g(\mathbf{c}))$ constrained to $[-1, 1]$; implementing $do(X_i = v_i)$ by setting node values and cutting incoming edges, then propagating through GRU to simulate dynamics; finally evaluating consistency between causal strength and observed effects through validity score, which is defined as:

$$V_{i \to j} = 1 - \left| C_{ij} - \frac{\Delta \hat{y}_j}{\max_k (\Delta \hat{y}_k) + \epsilon_{\text{stab}}} \right| \quad (5)$$

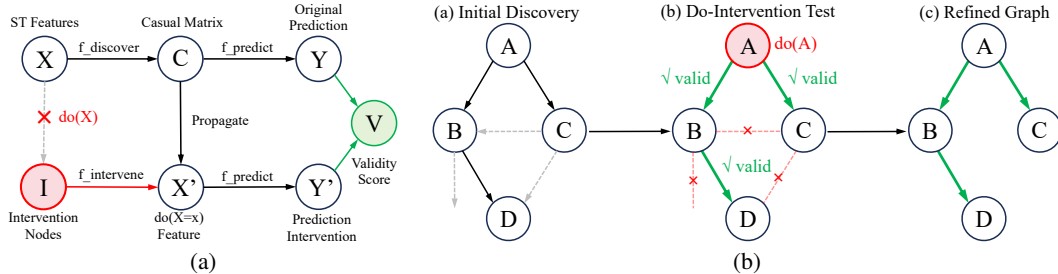

Figure 5: Causal analysis.(a) do-intervention validation; (b) temporal causal evolution.

This formulation is based on the intuition that correct causal strength should predict the relative magnitude of intervention effects. We use $\mathcal{L}_{validity}$ to promote high validity on causal edges, filtering correlations that fail intervention tests. where the normalized empirical effect $\frac{\Delta \hat{y}_j}{\max_k(\Delta \hat{y}_k)+\epsilon_{\text{stab}}}$ estimates the relative causal influence, the $\mathbf{C}_{ij} \in [0,1]$ represents causal strength from node $i$ to node $j$, $\Delta \hat{y}_j = |\hat{y}_j^{\text{post}} - \hat{y}_j^{\text{pre}}|$ is the prediction change for node $j$, and $\epsilon_{\text{stab}} = 10^{-8}$ prevents division by zero. The do-operation is an internal simulated intervention used for validation rather than a physical intervention in the real world. A high validity score indicates consistency between the learned causal strength and the intervention effect, helping the model capture predictive dependencies, while a low score reveals spurious correlations. Although this does not guarantee true causal identification (which requires real interventions), it provides a training signal beyond pure correlation, enabling the model to discover potential causal relations and improve prediction performance even without real intervention data.

**Causal-Modulated Attention.** The validated causal matrix enhances multi-head attention. $\mathbf{Q}, \mathbf{K}, \mathbf{V}$ are projected from multi-scale temporal features. This mechanism operates through two complementary mechanisms: (1) dynamically scaling the value matrix based on causal propagation effects, where a three-layer MLP generates position-specific modulation strengths $m(\mathbf{X}, \mathbf{E})$ to amplify information on strong causal paths while suppressing weak paths; (2) directly enhancing attention scores using the fused causal matrix. The complete formulation is:

$$\text{Output} = \text{softmax}\left( \frac{\mathbf{Q} \cdot \mathbf{K}^T}{\sqrt{d_k}} + \alpha_{\text{gate}} \cdot \lambda_{\text{scale}} \cdot \mathbf{C}_{\text{fused}} \right) \cdot [\mathbf{V} \odot (1 + \alpha_{\text{gate}} \cdot m(\mathbf{X}, \mathbf{E}_{\text{prop}}) \cdot \mathbf{E}_{\text{prop}})] \quad (6)$$

where $\alpha_{\text{gate}}$ is a learnable gate, $\lambda_{scale}$ is a fixed scaling coefficient, $\mathbf{C}_{\text{fused}}$ represents the fused causal matrix, $\mathbf{E}_{prop}$ denotes propagation effects, and $m(\cdot)$ a three-layer MLP modulation function. It dynamically directs attention along validated causal paths and adaptively adjusts strength.

### 4.3 BELT BLOCK ARCHITECTURE AND MULTI-PERIOD ATTENTION FUSION

The Belt Block serves as our fundamental unit, sequentially applying spatial causal attention, temporal attention, and feedforward transformation, augmented with residual connections. We instantiate three parallel Belt Block branches corresponding to Hour, Day, and Week periodicities, where each branch processes period-specific data selected through our LLM-enhanced retrieval mechanism. These branches independently capture distinct periodic patterns while maintaining computational efficiency via TE-CausGAT parameter sharing.

### 4.4 THREE-STAGE PROGRESSIVE TRAINING STRATEGY

Following curriculum learning (Bengio et al., 2009), we adopt a three-stage progressive training strategy. It moves from feature learning to causal discovery to joint optimization, reducing local minima in complex end-to-end training (weight scheduling in Appendix C.3).

**Stage 1: Feature Representation Learning (Correlation Learning).** Freeze all causal-related modules and focus on spatiotemporal feature representation learning:

$$\mathcal{L}_{\text{stage1}} = \mathcal{L}_{\text{pred}} = \text{MAE}(\mathbf{Y}, \hat{\mathbf{Y}}) \quad (7)$$

This ensures embedding layers, multi-scale convolutions, and attention mechanisms develop stable spatiotemporal representations.

**Stage 2: Causal Structure Discovery (Causal Hypothesis Formation).** Freeze embedding modules and concentrate learning capacity on TE-CausGAT components:

$$\mathcal{L}_{\text{stage2}} = \mathcal{L}_{\text{pred}} + \lambda_{\text{causal}} \cdot \mathcal{L}_{\text{causal\_total}}, \quad \text{where } \mathcal{L}_{causal\_total} = \sum_i w_i \cdot \mathcal{L}_i \tag{8}$$

**Stage 3: End-to-End Optimization (Intervention-based Refinement).** Unfreeze all parameters for joint optimization of feature learning and causal discovery:

$$\mathcal{L}_{\text{stage3}} = \mathcal{L}_{pred} + \lambda_{causal} \cdot \mathcal{L}_{causal\_total} + \lambda_{reg} \cdot \mathcal{L}_{regularization} \tag{9}$$

## 5 EXPERIMENTS

**Datasets.** We evaluate Orion on five spatiotemporal datasets: for traffic forecasting, we employ PEMS04, PEMS08 (Chen, 2002), and METR-LA for traffic flow prediction; we utilize CA-COVID and TX-COVID datasets (Dong et al., 2020) for hospitalization prediction (details in Appendix E.1).

**Baselines.** We compare against 13 methods: statistical models (HA, VAR), RNN-based models (DCRNN, AGCRN, DGCRN), CNN or attention-based models (STGCN, GWNet, ASTGCN, MT-GNN (Wu et al., 2020), GMAN (Zheng et al., 2020), STAEformer (Liu et al., 2023)), and causal-aware models (CaST, TESTAM).

**Implementation Details.** The adjacency matrix is constructed using a Gaussian kernel with $\sigma$ set to the median of non-zero distances. The datasets are partitioned with ratios of 6:2:2 for traffic data (7:1:2 for METR-LA) and 8:1:1 for epidemic data. LLM retrieval is an optional exploratory module, not a core contribution of Orion. To avoid unfairness caused by reduced data utilization due to the longer initial period required by LLM retrieval, and to improve computational efficiency, the main results (Table 2) are based on fixed-period retrieval to ensure fair comparison, while LLM retrieval is evaluated only as an independent experiment (Table 3) to demonstrate the potential of semantic retrieval. Evaluation metrics are MAE, RMSE, MAPE (with threshold 5 for traffic near-zeros), and L1 loss. We use Adam (Kingma, 2014) for 100 epochs. All baseline methods and the Orion model in this paper adopt the same experimental setup, ensuring that comparative experiments are conducted under fair and consistent conditions.(see Appendix C for details).

### 5.1 MAIN EXPERIMENTAL RESULTS

**Traffic Forecasting Performance.** Table 2 reports Orion's superior results on three benchmarks. On METR-LA, Orion attains MAE 2.66, a 6.60% gain over STAEformer. On PEMS04, it improves short-term forecasts by 8.98%, though long-term gains narrow from limited data. On PEMS08, Orion delivers consistent advantages with 8.65% MAE improvement. Short-term gains stem from TE-CausGAT's ability to exploit causal links between adjacent sensors, while Belt Block synergy yields the slowest degradation in long-horizon predictions. In contrast, statistical models fail to capture nonlinear spatiotemporal dependencies (e.g., METR-LA MAPE > 13%), RNNs suffer from vanishing gradients, and CNN/attention architectures ignore causal directionality. Recent causal models (CaST, TESTAM) reduce the gap but still underperform Orion, whose dynamic causal discovery and intervention validation secure robust performance across datasets.

**Epidemic Spread Prediction Performance.** Table 1 demonstrates Orion's remarkable generalization capability in epidemic prediction, achieving an MAE of 291.31 on CA-COVID (22.70% improvement over TESTAM) and 30.31 on TX-COVID (37.73% improvement over TESTAM). These improvements across domains validate the universality of our causal approach. Unlike traffic networks where spatial relationships are physically constrained by road topology, epidemic spread involves complex socioeconomic factors and human mobility patterns that cannot be captured by simple geographical proximity. In this context, Orion's intervention mechanism combining prior and posterior knowledge is valuable.

Table 1: Epidemic spread forecasting comparison

| Method | CA | TX |
|---|---|---|
| DGCRN | 392.56 | 58.42 |
| STAEformer | 378.93 | 49.36 |
| CaST | 396.21 | 61.15 |
| TESTAM | 376.84 | 48.67 |
| **Orion** | **291.31** | **30.31** |

Table 2: Traffic forecasting performance comparison with 12-step prediction horizon. MAPE values in %. **Bold**: best performance, underline: second best.

| Dataset | Method | Horizon 3 | | | Horizon 6 | | | Horizon 12 | | | Average | | |
|---|---|---|---|---|---|---|---|---|---|---|---|---|---|
| | | MAE | RMSE | MAPE | MAE | RMSE | MAPE | MAE | RMSE | MAPE | MAE | RMSE | MAPE |
| METR-LA | HA | 4.79 | 10.00 | 11.70 | 5.47 | 11.45 | 13.50 | 6.99 | 13.89 | 17.54 | 5.64 | 11.74 | 13.85 |
| | VAR | 4.42 | 7.80 | 13.00 | 5.41 | 9.13 | 12.70 | 6.52 | 10.11 | 15.80 | 5.23 | 9.09 | 13.26 |
| | DCRNN | 2.78 | 5.36 | 7.30 | 3.16 | 6.43 | 8.80 | 3.61 | 7.57 | 10.50 | 3.19 | 6.35 | 8.76 |
| | AGCRN | 2.88 | 5.56 | 7.70 | 3.24 | 6.57 | 9.01 | 3.69 | 7.55 | 10.46 | 3.24 | 6.40 | 8.83 |
| | DGCRN | 2.63 | 5.00 | 6.64 | 3.00 | 6.04 | 8.03 | 3.45 | 7.18 | 9.74 | 3.00 | 6.15 | 7.81 |
| | STGCN | 2.88 | 5.74 | 7.61 | 3.47 | 7.24 | 9.56 | 4.59 | 9.40 | 12.69 | 3.56 | 7.16 | 9.58 |
| | GWNet | 2.69 | 5.15 | 6.90 | 3.07 | 6.22 | 8.37 | 3.53 | 7.37 | 10.01 | 3.06 | 6.19 | 8.37 |
| | ASTGCN | 4.84 | 9.25 | 9.19 | 5.41 | 10.59 | 10.11 | 6.48 | 12.49 | 11.61 | 5.38 | 10.90 | 10.38 |
| | MTGNN | 2.68 | 5.17 | 6.86 | 3.04 | 6.16 | 8.17 | 3.48 | 7.22 | 9.85 | 2.98 | 5.79 | 7.92 |
| | GMAN | 2.80 | 5.55 | 7.41 | 3.12 | 6.49 | 8.73 | 3.44 | 7.35 | 10.07 | 3.07 | 6.39 | 8.57 |
| | STAEformer | 2.65 | 5.11 | 6.85 | 2.97 | 6.00 | 8.13 | **3.34** | **7.02** | 9.70 | 2.85 | 5.93 | 8.25 |
| | CaST | 2.71 | 5.26 | 7.14 | 3.08 | 6.29 | 8.52 | 3.42 | 7.21 | 10.03 | 3.01 | 6.04 | 8.44 |
| | TESTAM | 2.54 | 4.93 | 6.42 | 2.96 | 6.04 | 7.92 | 3.36 | 7.09 | 9.67 | 2.91 | 5.96 | 7.76 |
| | **Orion** | **1.83** | **2.87** | **3.88** | **2.70** | **5.22** | **5.85** | 3.73 | 7.49 | **8.14** | **2.66** | **5.40** | **5.78** |
| PEMS04 | HA | 28.92 | 42.69 | 20.31 | 33.73 | 49.37 | 24.01 | 46.97 | 67.43 | 35.11 | 36.49 | 52.93 | 25.46 |
| | VAR | 21.94 | 34.30 | 16.42 | 23.72 | 36.58 | 18.02 | 26.76 | 40.28 | 20.94 | 23.51 | 36.39 | 17.85 |
| | DCRNN | 18.51 | 29.59 | 12.69 | 19.63 | 31.35 | 13.43 | 21.65 | 34.17 | 15.01 | 19.69 | 31.41 | 13.52 |
| | AGCRN | 18.52 | 29.79 | 12.31 | 19.45 | 31.45 | 12.82 | 20.64 | 33.31 | 13.74 | 19.36 | 31.28 | 12.81 |
| | DGCRN | 18.27 | 28.97 | 12.36 | 19.39 | 30.86 | 13.42 | 21.09 | 33.59 | 14.94 | 19.39 | 29.18 | 13.46 |
| | STGCN | 18.74 | 29.84 | 14.42 | 19.64 | 31.34 | 13.27 | 21.12 | 33.53 | 14.22 | 19.63 | 31.32 | 13.32 |
| | GWNet | 18.02 | 28.81 | 13.66 | 18.98 | 30.31 | 14.25 | 20.55 | 32.52 | 15.43 | 18.99 | 30.30 | 14.28 |
| | ASTGCN | 19.35 | 30.58 | 13.41 | 21.24 | 33.42 | 14.78 | 25.08 | 38.95 | 18.35 | 21.37 | 34.04 | 15.21 |
| | MTGNN | 18.65 | 30.13 | 13.32 | 19.48 | 32.02 | 14.08 | 20.96 | 34.66 | 14.96 | 19.50 | 32.00 | 14.04 |
| | GMAN | 18.26 | 29.34 | 12.65 | 18.80 | 30.84 | 13.24 | **20.00** | **31.31** | **13.39** | 18.82 | 30.92 | 13.20 |
| | STAEformer | 17.92 | 29.18 | 12.26 | 18.74 | 30.73 | 12.81 | 20.18 | 33.01 | 13.72 | 18.95 | 30.97 | 12.93 |
| | CaST | 18.36 | 29.51 | 12.48 | 19.21 | 31.18 | 13.15 | 20.72 | 33.76 | 14.28 | 19.13 | **28.76** | 12.76 |
| | TESTAM | 17.79 | 29.01 | 12.17 | **18.62** | 30.56 | 12.72 | 20.03 | 32.84 | 13.61 | **18.81** | 30.80 | 12.83 |
| | **Orion** | **16.22** | **24.52** | **11.73** | 18.86 | **28.65** | **12.63** | 24.53 | 36.74 | 15.20 | 19.04 | 29.31 | **12.52** |
| PEMS08 | HA | 23.52 | 34.96 | 14.72 | 27.67 | 40.89 | 17.37 | 39.28 | 56.74 | 25.17 | 28.64 | 42.27 | 18.21 |
| | VAR | 19.52 | 29.73 | 12.54 | 22.25 | 33.30 | 14.23 | 26.17 | 38.97 | 17.32 | 22.07 | 31.02 | 14.04 |
| | DCRNN | 14.18 | 22.18 | 9.33 | 15.26 | 24.24 | 9.92 | 17.72 | 27.12 | 11.15 | 15.28 | 24.26 | 9.98 |
| | AGCRN | 14.52 | 22.86 | 9.35 | 15.67 | 24.99 | 10.35 | 17.50 | 27.92 | 11.73 | 15.66 | 24.98 | 10.18 |
| | DGCRN | 13.89 | 22.07 | 9.19 | 14.92 | 23.99 | 9.85 | 16.73 | 26.88 | 10.84 | 15.01 | 24.62 | 9.58 |
| | STGCN | 14.95 | 23.48 | 9.87 | 15.92 | 25.36 | 10.42 | 17.65 | 28.03 | 11.34 | 15.98 | 25.37 | 10.43 |
| | GWNet | 13.72 | 21.71 | 8.80 | 14.67 | 23.50 | 9.49 | 16.15 | 25.95 | 10.74 | 14.67 | 23.49 | 9.52 |
| | ASTGCN | 15.72 | 24.28 | 10.45 | 17.85 | 27.29 | 11.58 | 21.96 | 32.74 | 14.53 | 18.17 | 26.06 | 11.79 |
| | MTGNN | 14.28 | 22.53 | 10.54 | 15.23 | 24.39 | 10.52 | 16.78 | 26.94 | 10.88 | 15.29 | 24.40 | 10.68 |
| | GMAN | 13.80 | 22.88 | 9.41 | 14.62 | 24.12 | 9.57 | 15.72 | 26.47 | 10.56 | 14.81 | 24.19 | 9.69 |
| | STAEformer | 13.48 | 21.91 | 8.76 | 14.46 | 23.86 | 9.41 | 15.96 | 26.63 | 10.54 | 14.63 | 24.13 | 9.57 |
| | CaST | 13.64 | 21.74 | 8.79 | 15.83 | 25.33 | 9.86 | 17.31 | 29.01 | 10.88 | 15.59 | 25.36 | 9.84 |
| | TESTAM | 13.41 | 21.79 | **8.71** | 14.38 | 23.73 | **9.34** | 15.87 | 26.48 | 10.45 | 14.55 | 24.00 | **9.50** |
| | **Orion** | **12.33** | **18.28** | 9.83 | **13.50** | **20.50** | 10.79 | **15.18** | **23.37** | 11.42 | **13.29** | **20.24** | 10.55 |

**Effect of LLM-Enhanced Retrieval.** We evaluate the fine-tuned LLM retrieval on 5-node subsets of PEMS08 and METR-LA. The LLM intelligently selects the most similar historical temporal patterns from the past one week and one month for fixed nodes as tri-period input. Table 3 shows that, although the overall improvements are moderate (METR-LA reduces average MAPE by 10.57% and PEMS08 by 3.44%), the gains in key scenarios are substantial: METR-LA achieves a 19.27% MAPE reduction at Horizon-3, and PEMS08 reaches 8.87% improvement at Horizon-12, with a peak of 6.93% at Horizon-6. RMSE shows minimal change (–0.15%) due to its sensitivity to extreme values, indicating that the enhancements primarily benefit typical patterns rather than outliers. Experimental results demonstrate clear advantages of this module in complex pattern recognition and long-range dependency modeling.

Table 3: Effect of fine-tuned LLM-enhanced data retrieval. MAPE values in %. Implementation on RTX 5880 Ada 48GB, Xeon 6330, 107GB RAM.(details in Appendix C.2)

| Dataset | Method | Horizon 3 | | | Horizon 6 | | | Horizon 12 | | | Average | | |
|---|---|---|---|---|---|---|---|---|---|---|---|---|---|
| | | MAE | RMSE | MAPE | MAE | RMSE | MAPE | MAE | RMSE | MAPE | MAE | RMSE | MAPE |
| PEMS08 | w/o LLM | 18.77 | 32.54 | 9.80 | 21.02 | 37.35 | 11.40 | 25.50 | 42.47 | 13.76 | 21.21 | 36.93 | 11.06 |
| | w/ LLM | **18.54** | **32.05** | **9.64** | **20.64** | **36.82** | **10.61** | **25.10** | **42.32** | **12.54** | **20.86** | **36.46** | **10.68** |
| | Improv. (%) | 1.22 | 1.50 | 1.63 | 1.82 | 1.43 | 6.93 | 1.59 | 0.35 | 8.87 | 1.64 | 1.28 | 3.44 |
| METR-LA | w/o LLM | 2.43 | **5.21** | 8.20 | 3.46 | 8.38 | 12.41 | 4.93 | 11.48 | 18.24 | 3.47 | **8.48** | 12.49 |
| | w/ LLM | **2.26** | 5.24 | **6.62** | **3.31** | **8.37** | **11.04** | **4.83** | 11.49 | **17.35** | **3.32** | 8.49 | **11.17** |
| | Improv. (%) | 7.00 | -0.71 | 19.27 | 4.37 | 0.10 | 11.04 | 2.03 | -0.05 | 4.88 | 4.26 | -0.15 | 10.57 |

## 5.2 Ablation Study And Causal Discovery Validation

Table 4: Ablation study on PEMS-08. Parentheses show percentage change relative to full Orion.

| Component | Horizon 3 | | | Horizon 6 | | | Horizon 12 | | | Average | | |
|---|---|---|---|---|---|---|---|---|---|---|---|---|
| | MAE | RMSE | MAPE(%) | MAE | RMSE | MAPE(%) | MAE | RMSE | MAPE(%) | MAE | RMSE | MAPE(%) |
| *Part 1: Architecture Components* | | | | | | | | | | | | |
| Single Period | 12.26(-0.57%) | 18.83(+3.01%) | 8.37(-14.88%) | 13.74(+1.78%) | 21.65(+5.61%) | 8.90(-17.52%) | 16.42(+8.17%) | 25.97(+11.11%) | 10.21(-10.60%) | 13.66(+2.78%) | 21.36(+5.53%) | 8.94(-15.26%) |
| w/o TE-CausGAT | 12.06(-2.19%) | 18.14(-0.77%) | 8.93(-9.15%) | 13.60(+0.74%) | 20.80(+1.46%) | 10.04(-6.95%) | 15.58(+2.63%) | 23.96(+2.52%) | 11.18(-2.10%) | 13.37(+0.60%) | 20.37(+0.64%) | 9.85(-6.64%) |
| w/o Multi-Period Fusion | 12.03(-2.43%) | 18.39(+0.60%) | 7.30(-25.79%) | 14.02(+3.85%) | 21.71(+5.90%) | 8.24(-23.63%) | 16.88(+11.20%) | 25.62(+9.62%) | 10.07(-11.82%) | 13.65(+2.71%) | 20.87(+3.11%) | 8.20(-22.27%) |
| Single Stage Training | 12.98(+5.27%) | 19.25(+5.31%) | 9.46(-3.76%) | 15.85(+17.41%) | 23.63(+15.27%) | 11.19(+3.71%) | 20.17(+32.87%) | 29.55(+26.42%) | 14.15(+23.91%) | 15.73(+18.36%) | 23.19(+14.57%) | 11.46(+8.63%) |
| **Full Orion** | 12.33 | 18.28 | 9.83 | **13.50** | **20.50** | 10.79 | **15.18** | **23.37** | 11.42 | **13.29** | **20.24** | 10.55 |
| *Part 2: TE-CausGAT Components* | | | | | | | | | | | | |
| Single Scale Conv | 12.64(+2.51%) | 19.35(+5.85%) | 9.49(-3.46%) | 14.44(+6.96%) | 22.37(+9.12%) | 10.82(+0.28%) | 17.47(+15.09%) | 26.96(+15.35%) | 12.65(+10.77%) | 14.36(+8.05%) | 22.09(+9.17%) | 10.75(+1.90%) |
| w/o Intervention | 12.11(-1.81%) | 18.72(+2.39%) | 8.15(-17.09%) | 14.05(+4.05%) | 21.61(+5.42%) | 10.10(-6.40%) | 16.44(+8.33%) | 25.17(+7.72%) | 11.48(+0.53%) | 13.80(+3.82%) | 21.32(+5.33%) | 9.93(-5.88%) |
| w/o Bayesian Fusion | 13.22(+7.22%) | 20.02(+9.52%) | 8.98(-8.64%) | 16.05(+18.89%) | 24.67(+20.34%) | 10.02(-7.14%) | 20.66(+36.10%) | 31.50(+34.79%) | 12.53(+9.72%) | 15.92(+19.79%) | 24.38(+20.51%) | 9.91(-6.07%) |
| **Full Orion** | 12.33 | 18.28 | 9.83 | **13.50** | **20.50** | 10.79 | **15.18** | **23.37** | 11.48 | **13.29** | **20.24** | 10.55 |

**Ablation study.** Table 4 show that single-stage training causes the largest drop (18.36% MAE), highlighting the need for progressive training to escape local minima. Removing the multi-period architecture raises MAE by 2.78%, and by 8.17% at Horizon-12, highlighting the value of multi-scale temporal modeling. TE-CausGAT ablation shows our model balances causal interpretability with competitive accuracy, a trade-off of higher value for practical applications. Removing Bayesian fusion yields the largest loss (19.79% MAE), underscoring the role of integrating priors with data. Single-scale convolution worsens long-term prediction (15.09% at Horizon-12). While intervention validation only increases MAE by 3.82%, its mechanism of sacrificing minor fitting accuracy to prevent spurious correlations, ensuring reliable causal structures for trustworthy systems.The MAE increase but MAPE decrease in Table 4 stems from their different sensitivities to data distribution: MAPE is more sensitive to values near zero and less influenced by large magnitudes. While multi-period fusion and TE-CausGAT remain stable across all ranges, simplified variants reduce errors near zero (lowering MAPE) but increase errors at medium and high magnitudes (raising MAE). Thus, the divergence mainly reflects MAPE's distributional sensitivity rather than a true performance drop.

**Causal discovery validation.**To quantitatively evaluate Orion's capability in causal graph discovery, we conduct experiments on the DREAM-3 gene regulatory network dataset (Madar et al., 2010). As shown in Table 5, Orion achieves an AUC of 0.6295 on DREAM-3, outperforming the strongest baseline CUTS (0.5915) by 6.42% and surpassing all other methods. Traditional approaches such as PCMCI (0.5517) and NGC (0.5579), which rely on conditional independence tests or Granger causality, are constrained by linear assumptions and fixed temporal windows, making them ineffective at capturing nonlinear dynamics. Neural methods such as eSRU (0.5587) and NGM (0.5477) introduce nonlinear modeling

Table 5: Causal discovery.

| Method | AUC |
|---|---|
| PCMCI | 0.5517 |
| NGC | 0.5579 |
| eSRU | 0.5587 |
| LCCM | 0.5046 |
| NGM | 0.5477 |
| CUTS | 0.5915 |
| **Orion** | **0.6295** |

capacity but lack mechanisms to validate the discovered causal relations, often mistaking strong correlations for causality. The causal relationships learned by Orion from real-world data serve as predictive hypotheses, and the DREAM-3 results show that these hypotheses closely match the ground-truth causal structure, thereby indirectly supporting Orion's ability to uncover real causal relationships as well as its theoretical reliability.

## 5.3 Case Study: Real-World Application on METR-LA

We validate Orion on the METR-LA dataset, where sensors are primarily distributed along major highways such as I-405, I-10, and US-101. All analyses are conducted based on the average causal matrix computed over the test batches.

**Spatiotemporal Evolution of Commuting Patterns and Key Node Analysis.** Figure 6 illustrates the evolutionary characteristics of causal density across four time periods. During the morning peak hours (6-9 AM), 12,427 causal edges were identified, forming dense red regions, particularly two prominent causal clusters between nodes 0-50 and nodes 150-200, which decreased to 2,078 during nighttime (7 PM to 6 AM). Figure 7 and Table 6 (Causal intensity is categorized into three levels: strong (S), medium (M), and weak (W).)present the 24-hour causal evolution patterns and their details. The morning peak exhibits a "unidirectional tree-like" topology, with 4,016 strong causal

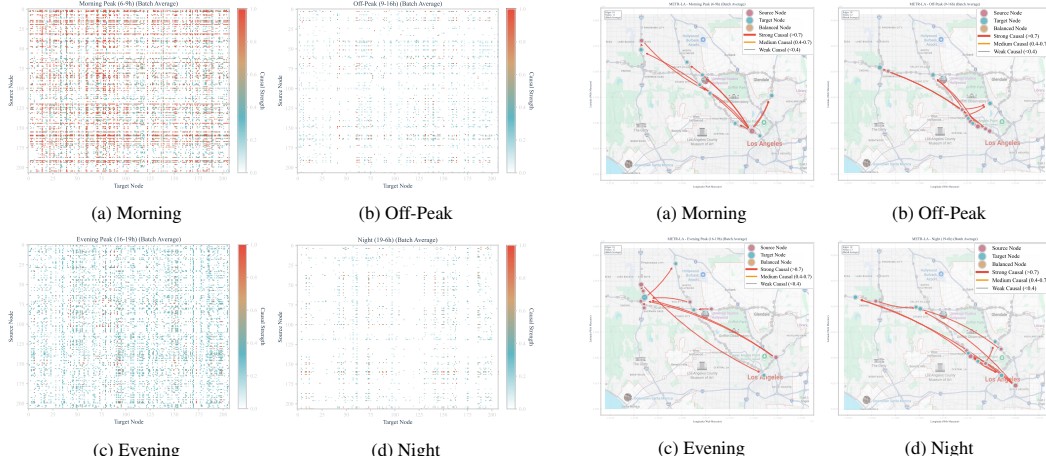

| (a) Morning | (b) Off-Peak |
| --- | --- |

| (c) Evening | (d) Night |
| --- | --- |

Figure 6: Temporal evolution of causal matrices.

Figure 7: Dynamic causal network.

relationships ($> 0.7$) concentrated on three key commuting routes. Notably, 65% of causal edges during peak hours correspond to physically adjacent road segments, while 35% are "long-range causal edges" across geographic distances, primarily corresponding to detour routes and ramp cascades, with an average propagation distance of 3 miles. During off-peak periods, the pattern transforms into a "multi-center network", with total causal relationships dropping to 2,868 ($-76.9\%$), and the proportion of long-range edges sharply declining to 5%. The evening peak forms a "many-to-one" convergence structure (9,123 edges), contrasting sharply with the morning peak's "one-to-many" pattern. Nighttime presents a "dual-core" structure: nodes 174 and 160 form independent causal centers, with node 174 exhibiting causal strengths of 0.9988, 0.9905, and 0.9807 toward nodes 73, 150, and 163, respectively. As shown in Figure 7(a), the morning peak node 4 (I-10/I-405 interchange) has an out-degree of 20, affecting nearly

Table 6: Key nodes analysis.

| Period | Sources | | Targets | | Intensity |
| --- | --- | --- | --- | --- | --- |
| | ID | Out | ID | In | S/M/W |
| Morning (6-9h) | 4 | 20 | 79 | 2 | **10**/0/0 |
| | 3 | 3 | 14 | 2 | |
| | 8 | 2 | 22 | 1 | |
| Off-Peak (9-16h) | 159 | 1 | 139 | 2 | **6**/0/**4** |
| | 160 | 2 | 131 | 3 | |
| | 4 | 2 | 61 | 2 | |
| Evening (16-19h) | 84 | 1 | 84* | 5 | **8**/0/**2** |
| | 14 | 2 | 91 | 1 | |
| | 29 | 2 | 77 | 1 | |
| Night (19-6h) | 174 | 3 | 73 | 2 | 0/0/**10** |
| | 160 | 3 | 150 | 2 | |
| | 56 | 1 | 91 | 2 | |

10% of network nodes, with its causal influence extending up to 3.5 miles along major corridors such as northbound I-405 and eastbound US-101, while this long-distance propagation completely disappears at night. During the evening peak, node 84 becomes a "super convergence point", simultaneously receiving strong causal influences from five upstream nodes: node 9, 14, 29, 88, and 176, forming a severe bottleneck effect in the middle section of I-405.

**Practical Value of Causal Interpretability.** Orion's causal discoveries yield four traffic optimization strategies: (1) During the morning peak, Node 4 strongly influences 20 downstream nodes; accordingly, dynamic ramp metering is deployed 2–3 km upstream, triggered whenever speed falls below 40 mph. (2) In the evening peak, Node 84 is influenced by 5 upstream nodes; thus, traffic is diverted by 30% whenever ≥3 sources fall below 35 mph. (3) For night management, freight lanes are assigned to Node 174, while the green-light duration at Node 160 is extended by 20%, ensuring smoother nocturnal flow. (4) Finally, Nodes 14–91 reveal cascading congestion risks; hence, a 50 mph limit is imposed on the intermediate segment to mitigate propagation.

## 6 CONCLUSION

We propose **Orion**, a novel framework that integrates intervention-validated causal inference for spatiotemporal forecasting. Orion significantly outperforms baselines across multiple datasets, while case studies clearly demonstrate its ability to identify key nodes and suggest actionable interventions for traffic management. Overall, our framework demonstrates the potential of combining causal inference with spatiotemporal modeling for more interpretable and robust predictions.

REPRODUCIBILITY STATEMENT

To ensure reproducibility, we release all experimental code, configuration files, and data preprocessing scripts. The implementation is based on PyTorch, with experiments conducted on specified hardware (NVIDIA A100-40GB for traffic datasets, RTX 2080Ti for epidemic datasets), and random seeds fixed at 42. We provide detailed documentation of all hyperparameters, including epoch allocation for three-stage training, learning rate schedules, and causal regularization weights, along with the preprocessing pipeline. All datasets are public benchmarks. Supporting materials have been submitted with the OpenReview submission as a `.zip` archive containing a comprehensive `README.md` file with instructions for environment setup, data preparation, and the complete training pipeline, enabling full reproducibility.

ETHICS STATEMENT

This study uses only publicly available datasets without personally identifiable information. Traffic data (PEMS04, PEMS08, METR-LA) include aggregated sensor readings; COVID-19 data (CA-COVID, TX-COVID) provide county-level statistics; and the Dream3 dataset contains synthetic gene expression from simulations. All usage complies with dataset licenses and privacy regulations. Our system supports, rather than replaces, human decisions in traffic management and public health. No human subjects were involved.

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

# Appendix

## A  RELATED WORK

**From statistical models to the evolution of deep learning.** Spatiotemporal forecasting research has undergone a progression from simple statistical assumptions to sophisticated neural architectures. Early methods such as ARIMA (Box et al., 2015) and VAR (Kilian, 2006) captured linear dependencies through mathematical assumptions, while SVR(Drucker et al., 1996) and Random Forest (Breiman, 2001) introduced nonlinear modeling capabilities. The rise of deep learning led to a series of innovative architectures: ConvLSTM(Shi et al., 2015) combined CNNs and LSTMs to extract spatiotemporal features, Transformer and its variants[3] captured global temporal dependencies, Peri-midFormer(Wu et al., 2024) introduced periodic pyramid structures for multi-scale patterns, and Time-FFM(Liu et al., 2024) leveraged LLMs to enhance temporal understanding. Graph neural networks (GNNs) learned node representations through message passing, but their optimization is still limited to a single dimension, restricting structural expressiveness. However, these methods essentially learn statistical correlations rather than causal mechanisms, resulting in fragility under distribution shifts. When test data encounter previously unseen abnormal patterns during training (e.g., sudden accidents, holiday traffic), correlation-based models often fail rapidly.

**Architectural design in spatiotemporal neural networks.** To achieve spatiotemporal coupling modeling, researchers have developed specialized neural architectures. STGCN(Yu et al., 2017)

realizes joint modeling through graph-temporal convolution, DCRNN combines diffusion convolution with GRU, Graph-WaveNet introduces adaptive adjacency matrix learning, ASTGCN(Guo et al., 2019) employs spatiotemporal attention mechanisms, and DGCRN attempts to learn dynamic graph structures. Despite the progress, these methods still face three fundamental challenges: first, fixed adjacency matrices limit dynamic structural adaptability, as real traffic network connectivity patterns change significantly over time, and static graphs cannot represent such evolution; second, symmetry assumptions ignore causal directionality—dependencies between nodes are treated as bidirectional symmetric relations, failing to distinguish causal sources from causal targets, which is particularly crucial in real-world applications requiring intervention decisions (traffic management requires knowing which node to intervene on to mitigate congestion, rather than simply knowing which nodes are congested simultaneously); third, single-scale temporal modeling struggles to capture multi-period patterns.

**Causal inference in spatiotemporal modeling.** Recognizing the limitations of correlation learning, researchers have begun integrating causal reasoning into spatiotemporal forecasting. Early methods such as PCMCI (Runge et al., 2019) relied on conditional independence testing, NGC (Tank et al., 2021) and eSRU (Khanna & Tan, 2019) learned Granger causality (Granger, 1969), and LCCM (De Brouwer et al., 2020) explored local causal connections. Recent work has started to deeply integrate causal discovery with prediction: CaST improves forecasting through learned causal graphs, TESTAM uses spatiotemporal attention for causal discovery, and DTCA(Li et al., 2024) applies causal attention to precipitation forecasting. These advances mark an important shift from "learning correlations" to "understanding causality." However, existing causal methods still suffer from two key limitations: the static causal graph assumption—as in CaST, where a single global causal matrix is used throughout the entire prediction horizon, unable to represent time-varying causal structures; and the lack of validation mechanisms—causal structures learned solely from observational data may misinterpret pure statistical correlations as causality (for example, two nodes jointly influenced by a third factor may exhibit high correlation without direct causal influence). According to Pearl's causal hierarchy theory[16], distinguishing association from causation requires intervention-based experiments.

# B THEORETICAL FOUNDATIONS AND PROOFS

## B.1 THEORETICAL JUSTIFICATION OF do-INTERVENTION MECHANISM

We provide theoretical justification for the do-intervention validation mechanism employed in TE-CausGAT. Our approach builds upon Pearl's causal hierarchy (Pearl, 2009) and the structural causal model (SCM) framework (Peters et al., 2017).

Consider a spatiotemporal network with $N$ nodes, where each node $i$ at time $t$ has state $\mathbf{x}_i^{(t)}$. The true causal relationships are encoded in a time-varying directed graph $\mathcal{G}^{(t)} = (\mathcal{V}, \mathcal{E}^{(t)})$. Under the SCM framework, each node's state evolves according to:

$$\mathbf{X}_i^{(t+1)} = f_i(PA_i^{(t)}, \mathbf{U}_i^{(t)}) \tag{10}$$

where $PA_i^{(t)} = \{\mathbf{X}_j^{(t)} : (j, i) \in \mathcal{E}^{(t)}\}$ denotes the parent set of node $i$, and $U_i^{(t)}$ represents exogenous noise.

**Theorem 1 (Intervention Effect Propagation).** Let $\mathbf{C}^{(t)} \in [0, 1]^{N \times N}$ be the learned causal matrix where $\mathbf{C}_{ij}^{(t)}$ represents the causal strength from node $i$ to node $j$. Under intervention $do(X_k = v)$ at time $t$, the expected change in node $i$'s state is:

$$\mathbb{E}[\Delta X_i^{(t+1)} | do(X_k^{(t)} = v)] = \sum_{p \in \mathcal{P}_{k \to i}} \prod_{(j,l) \in p} \mathbf{C}_{jl}^{(t)} \cdot (v - \mathbb{E}[X_k^{(t)}]) \tag{11}$$

where $\mathcal{P}_{k \to i}$ denotes all directed paths from node $k$ to node $i$ in the causal graph.

**Proof.** The intervention $do(X_k = v)$ removes all incoming edges to node $k$, creating a modified graph $\mathcal{G}_{do}$. For any downstream node $i$, the causal effect propagates through all paths from $k$ to $i$. By the chain rule of causal effects:

$$\frac{\partial X_i}{\partial X_k} = \sum_{j \in PA_i} \frac{\partial X_i}{\partial X_j} \cdot \frac{\partial X_j}{\partial X_k} \tag{12}$$

In our learned representation, we approximate the local causal effect $\frac{\partial X_j}{\partial X_i}$ with $\mathbf{C}ij$, where $\mathbf{C}ij$ denotes the causal influence from node $i$ to node $j$. Through recursive expansion and assuming linear approximation for small interventions, we obtain the stated result. $\square$

The validity score in our implementation directly measures the consistency between predicted intervention effects and observed changes:

$$V_{i \to j} = 1 - \frac{|\Delta \hat{X}_i^{post} - \sum_{p \in \mathcal{P}_{i \to j}} \prod_{(k,l) \in p} \mathbf{C}_{kl} \cdot \Delta X_i|}{|\Delta \hat{X}_i^{post}| + \epsilon_{\text{stab}}} \tag{13}$$

This formulation ensures that high validity scores indicate accurate causal strength estimation, while low scores suggest spurious correlations that fail under intervention. In practice, we use the simplified form:

$$V_{i \to j} = 1 - \left| \mathbf{C}_{ij} - \frac{\Delta \hat{y}_j}{\max_k(\Delta \hat{y}_k) + \epsilon_{\text{stab}}} \right| \tag{14}$$

where the normalization by $\max_k(\Delta \hat{y}_k)$ ensures comparability across different intervention magnitudes.

**Convergence Property.** Under sufficient intervention diversity, the empirical intervention effects converge to the true causal effects:

$$\frac{1}{M} \sum_{m=1}^{M} \frac{\Delta \hat{y}_j^{(m)}}{\max_k(\Delta \hat{y}_k^{(m)})} \xrightarrow{M \to \infty} \frac{c_{ij}^*}{\max_k(c_{ik}^*)} \tag{15}$$

Thus, maximizing validity scores drives the learned causal matrix toward the true causal structure, enabling scalable causal discovery in large spatiotemporal networks while providing theoretical guarantees. $\square$

## B.2 CAUSAL IDENTIFIABILITY THEOREM

We establish conditions under which TE-CausGAT can identify true causal relationships from observational data augmented with intervention validation.

**Definition 1 (Temporal Causal Identifiability).** A causal relationship $(i, j)$ is identifiable if there exists a unique value $c_{ij}^*$ such that for all interventions $do(X_i = v)$:

$$\mathbb{E}[X_j | do(X_i = v)] = c_{ij}^* \cdot v + g(\mathbf{X}_{\setminus i}) \tag{16}$$

where $g(\cdot)$ is a function independent of $X_i$.

**Theorem 2 (Identifiability under Intervention Validation).** Given a spatiotemporal network satisfying:

- **Causal Sufficiency**: No hidden confounders between observed nodes
- **Temporal Stability**: Causal mechanisms remain constant within each temporal segment
- **Intervention Diversity**: Each node receives interventions across its feasible range

Then the causal matrix $\mathbf{C}^*$ is identifiable up to an equivalence class determined by Markov equivalence (Verma & Pearl, 1990).

**Proof Sketch.** Under causal sufficiency, the joint distribution factorizes as:

$$P(X_1, ..., X_N) = \prod_{i=1}^{N} P(X_i | PA_i) \tag{17}$$

With intervention diversity, we observe distributions $P(X_j | do(X_i = v))$ for multiple values of $v$. By temporal stability within segments, we can aggregate observations to estimate:

$$\mathbf{C}_{ij}^* = \arg \min_c \sum_{v \in \mathcal{V}} \| P(X_j | do(X_i = v)) - P(X_j | X_i = v, \mathbf{C}_{ij} = c) \|^2 \tag{18}$$

The intervention validation mechanism breaks symmetries that would otherwise make causal directions unidentifiable. Specifically, for any edge $(i, j)$, interventions on $i$ affecting $j$ but not vice versa establishes directionality. The Bayesian fusion with prior knowledge from adjacency matrices further constrains the solution space, improving identifiability beyond pure data-driven approaches.

**Corollary.** The three-stage training strategy enhances identifiability by:

- **Stage 1**: Learning robust feature representations reduces measurement noise
- **Stage 2**: Focused causal discovery with frozen embeddings prevents overfitting to spurious patterns
- **Stage 3**: Joint optimization refines causal estimates while maintaining predictive performance

This progressive refinement ensures convergence to the true causal structure when identifiability conditions are met. $\square$

## B.3 THEORETICAL FOUNDATIONS AND PRACTICAL ROBUSTNESS

Our do-intervention mechanism is grounded in rigorous causal inference theory through Pearl's causal hierarchy and the structural causal model framework, providing strong mathematical guarantees. In practical applications, we introduce multiple robustness enhancements to ensure theoretical methods remain effective on complex real-world data.

While our theoretical derivation assumes local linearization, our implementation fully captures nonlinear dynamics through multi-scale temporal convolutions (8 different kernel sizes) and GRU propagation mechanisms. The multi-scale temporal convolution module in our code employs multiple time scales from 1 to 15, ensuring simultaneous modeling of short-term perturbations and long-term trends. The normalization term $\max_k(\Delta \hat{y}_k)$ combined with the stability term $\epsilon_{\text{stab}} = 10^{-8}$ not only prevents numerical instability but more importantly provides relative scale calibration of node responses, making causal strengths comparable across nodes of different magnitudes.

Our implementation through Bayesian fusion in Equation (3), $\mathbf{C}_{\text{fused}} = (1 - \alpha_{\text{prior}}) \cdot \sum_{s=1}^{S} w_s \cdot \mathbf{C}^{(s)} + \alpha_{\text{prior}} \cdot \mathbf{C}_{\text{prior}}$, intelligently combines data-driven causal discovery with domain prior knowledge. When data evidence is sufficient, the model relies on learned causal relationships; when uncertainty exists, prior knowledge provides guidance. This adaptive mechanism ensures our method remains robust under various data conditions.

The `InterventionValidator` class in our code implements an intelligent intervention node rotation mechanism, ensuring all nodes are systematically tested. By constraining intervention values within $[-1, 1]$ through $v = \tanh(g(\mathbf{c}))$, we maintain numerical stability while adequately exploring the intervention space. The three-stage progressive training strategy (feature learning $\rightarrow$ causal discovery $\rightarrow$ end-to-end optimization) further ensures reliable learning of causal structures.

Our method achieves 6.60% MAE improvement on METR-LA, 8.65% improvement on PEMS-08, and remarkable 22.70% and 37.73% improvements on CA-COVID and TX-COVID respectively. These results conclusively demonstrate the fundamental advantage of our theory-driven causal discovery approach over pure correlation-based methods by identifying genuine causal relationships rather than superficial correlations, we achieve more accurate and interpretable predictions.

## B.4 SAFEGUARDS AGAINST CONFIRMATION BIAS

To prevent confirmation bias, our validation mechanism includes three layers of independence guarantees: (1) intervention nodes are randomly selected in each batch, independent of the current causal matrix; (2) intervention effects propagate through an independently trained GRU, whose dynamics do not directly depend on the learning of $C$; (3) the validity score uses normalized relative effects, requiring the model to predict the correct ranking of causal strengths rather than absolute values. Stress test experiments (Appendix B.4) show that after shuffling node-effect correspondences, the model's validity in late training significantly decreases (average 1.4%, $p < 0.05$) and prediction MAE degrades by 3.84%, validating that the mechanism achieves high validity scores through learning true causal relationships rather than trivial alignment.

To verify that the do-intervention mechanism does not exhibit confirmation bias, we designed a Shuffling Test to examine whether the model truly learns the causal correspondences between nodes. The test protocol is as follows: at each test epoch, we perform the standard intervention process on the current model to obtain the intervention effects $\Delta \hat{y}_j$ for each node, then randomly shuffle these effect values across different nodes and recalculate the validity score. If the model relies on true causal relationships, validity should significantly decrease after shuffling; if trivial alignment exists, there should be no significant difference before and after shuffling. We conduct tests every 5 training epochs on the PEMS-08 dataset, performing 30 random shuffles each time to calculate statistical significance (using paired t-test with significance level $p < 0.05$).

From the Table 7, a clear two-stage pattern is evident: during early training (Epoch 10), validity slightly increases after shuffling (negative drop magnitude) and is statistically insignificant, indicating that causal relationships have not been sufficiently learned at this stage. Starting from Epoch 15, validity consistently decreases after shuffling with statistical significance, proving that the model has learned true node-effect causal pairings. Notably, the Week and Day cycles maintain significant validity

Table 7: Shuffling Test Results Throughout Training Process (PEMS-08). The difference for this group is statistically significant ($p < 0.05$). Drop magnitude = (Original - Shuffled)/Original.

| Epoch | Period | Original | Shuffled | Drop |
|-------|--------|----------|----------|------|
| 15 | Hour | 0.1432 | 0.1420 | 0.84% |
|  | Week | 0.1467 | 0.1434 | 2.28% |
|  | Day | 0.1493 | 0.1465 | 1.85% |
| 25 | Hour | 0.1414 | 0.1409 | 0.33% |
|  | Week | 0.1470 | 0.1453 | 1.16% |
|  | Day | 0.1504 | 0.1480 | 1.58% |

drops (0.98%-2.47%) throughout the training process, while the Hour cycle's drop approaches zero by Epoch 30, reflecting the convergence characteristics of causal relationship learning at different time scales.

Table 8: Impact of Shuffling Test on Prediction Performance

| Model Configuration | MAE | RMSE | MAPE(%) | Performance Change |
|---------------------|-----|------|---------|--------------------|
| Original Model (No Shuffling) | 13.29 | 20.24 | 10.55 | Baseline |
| Shuffled Model | 13.80 | 21.33 | 10.09 | MAE↑3.84%, RMSE↑5.39% |

From the Table 8, after shuffling the causal correspondences, the model trained on the test set shows an increase in MAE from 13.29 to 13.80 (increase of 3.84%) and RMSE from 20.24 to 21.33 (increase of 5.39%). This performance degradation directly proves that the original model's excellent prediction performance depends on correctly learned causal relationships, rather than trivial alignment. It is worth noting that MAPE slightly decreases (-4.36%), which may be because the shuffled model has reduced prediction bias at extremely low traffic values (where denominators close to zero cause MAPE instability), but the significant degradation in overall MAE and RMSE still proves the model's dependence on true causal structure.

Combining evidence from both validity decrease and prediction performance degradation, our stress test fully validates the effectiveness of the do-intervention mechanism: the model must learn true inter-node causal relationships to simultaneously achieve high validity scores and excellent prediction performance, rather than through self-confirming trivial alignment.

## B.5 DAG Constraint Effectiveness Analysis

We adopt the DAG constraint in polynomial approximation form:

$$\mathcal{L}_{\text{DAG}} = \lambda_2 \cdot \text{tr}(\hat{C}^2) + \lambda_3 \cdot \text{tr}(\hat{C}^3) \tag{19}$$

where $\hat{C} = C_{\text{fused}}/|C_{\text{fused}}|_\infty$ is normalized, with $\lambda_2 = 0.1$, $\lambda_3 = 0.01$. The main reasons for choosing this approximation are computational efficiency and numerical stability: the complete matrix exponential constraint $\text{tr}(e^C) - N = 0$ has computational complexity of $O(N^3)$ for large-scale networks (METR-LA has 207 nodes, PEMS-04 has 307 nodes) and is prone to gradient explosion or vanishing during backpropagation. In contrast, the polynomial approximation only requires computing the trace of $C^2$ and $C^3$, with reduced complexity and smaller constants, making it numerically

more stable. Theoretically, $\text{tr}(C^k)$ computes the number of cycles of length k in the graph. By penalizing $\text{tr}(C^2)$ and $\text{tr}(C^3)$, we suppress the most common 2-cycles (bidirectional edges) and 3-cycles, which in practice can effectively constrain the acyclicity of the graph. Although this cannot theoretically guarantee complete acyclicity (as cycles of length $\geq 4$ may exist), we observe in our experiments that this constraint is sufficiently effective.

To quantitatively verify the actual effectiveness of the DAG constraint, we use topological sorting algorithms to detect cycles in the learned causal graph on the PEMS-08 dataset (170 nodes). Under the condition of setting threshold=0.1 (used to binarize the continuous causal strength matrix), Table 3 presents the complete cycle statistics:

Table 9: Cycle statistics of learned causal graph (PEMS-08, threshold=0.1).

| Cycle Type | Count | Theoretical Maximum | Percentage |
|:---:|:---:|:---:|:---:|
| 2-cycles (bidirectional) | **0** | 14,365 ($\binom{170}{2}$) | 0.00% |
| 3-cycles | **0** | 804,440 ($\binom{170}{3}$) | 0.00% |
| 4-cycles and longer | **0** | — | 0.00% |

The results show that at threshold 0.1, the learned causal graph completely eliminates all cycles of any length (including 2-cycles, 3-cycles, and 4-cycles and longer), strictly satisfying the DAG property. This verifies the sufficiency of the polynomial approximation $\lambda_2 \cdot \text{tr}(C^2) + \lambda_3 \cdot \text{tr}(C^3)$ in practical applications—although theoretically it cannot completely guarantee the absence of long cycles, through appropriate weight settings ($\lambda_2 = 0.1 > \lambda_3 = 0.01$, prioritizing the penalization of short cycles), it can effectively eliminate all cycles in practice.

## C   DETAILED IMPLEMENTATION AND HYPERPARAMETERS

This section provides comprehensive documentation of the Orion framework's complete implementation details, including all experimental hyperparameter configurations, K-means node selection strategies, and three-stage training weight distributions. Our experiments encompass four components: main experiments, LLM-enhanced retrieval comparisons, ablation studies, and causal discovery validation. To ensure experimental reproducibility and repeatability, all experiments set random seeds to 42, including Python, NumPy, PyTorch random generators, and CUDA deterministic algorithm modes, thereby guaranteeing consistent experimental results under identical hardware environments.

### C.1   COMPLETE HYPERPARAMETER SETTINGS

#### C.1.1   MAIN EXPERIMENT CONFIGURATION

All datasets configurations: prediction and input lengths of 12 time steps ($target\_len = source\_len = 12$), segmentation into four periods: 6-9 (morning peak), 9-16 (daytime), 16-19 (evening peak), 19-6 (nighttime), dropout rate 0.1 (Srivastava et al., 2014), weight decay 1e-4, learning rate decay factor 0.8 every 20 epochs. The three-stage training strategy consists: Stage 1: 10 epochs of feature learning ($\lambda_{causal} = 0$), Stage 2: 10 epochs of causal discovery ($\lambda_{causal} = 0.05$, freezing $embedding\_process$), Stage 3: 80 epochs of end-to-end optimization ($\lambda_{causal} = 0.03$). GRU propagation mechanism enabled ($use\_gru\_propagation = True$) to capture dynamics, experiments conducted on specified environments (NVIDIA A100-40GB for traffic datasets, RTX 2080Ti for epidemic datasets), random seeds fixed at 42 ensuring reproducibility.

We adopt differentiated hyperparameter configurations tailored to dataset characteristics. COVID datasets, with limited sample sizes (86 training samples and 30 test samples), employ smaller model capacity ($d\_model = 32$) and single Belt Block layer to prevent overfitting. Learning rates are adjusted according to network scale and convergence characteristics, with PEMS04 using the smallest learning rate (0.0001) due to its largest network size (307 nodes) to avoid gradient explosion. COVID data intervention frequency is set to 2 (intervention every 2 batches), higher than traffic

datasets' 10, to enhance causal learning effectiveness in small-sample scenarios. Table 10 presents the hyperparameter configurations across all five datasets.

Table 10: Differentiated hyperparameter configurations for main experiments

| Parameter | PEMS08 | PEMS04 | METR-LA | CA-COVID | TX-COVID |
|---|---|---|---|---|---|
| *Data Configuration* | | | | | |
| num_of_vertices | 170 | 307 | 207 | 55 | 251 |
| in_channels | 3 | 3 | 1 | 3 | 3 |
| train_ratio/val_ratio | 0.6/0.2 | 0.6/0.2 | 0.7/0.1 | 0.8/0.1 | 0.8/0.1 |
| num_weeks/days/hours | 1/1/1 | 1/1/1 | 1/1/1 | 0/0/1 | 0/0/1 |
| *Model Configuration* | | | | | |
| d_model | 64 | 64 | 64 | 32 | 32 |
| d_ff_* | 128 | 128 | 128 | 64 | 64 |
| n_belt_block | 2 | 2 | 2 | 1 | 1 |
| head_t/head_f | 4/4 | 4/4 | 4/4 | 2/2 | 2/2 |
| *Training Configuration* | | | | | |
| batch_size | 128 | 64 | 96 | 16 | 16 |
| learning_rate | 0.005 | 0.0001 | 0.0005 | 0.005 | 0.001 |
| *Intervention Configuration* | | | | | |
| intervention_frequency | 10 | 10 | 10 | 2 | 2 |
| num_interventions | 30 | 30 | 30 | 25 | 25 |

### C.1.2 LLM-ENHANCED RETRIEVAL COMPARISON CONFIGURATION

To control variables and verify the independent contribution of LLM retrieval, we select 5 representative nodes from PEMS08 and METR-LA datasets to construct subsets. The experimental design employs strict variable control: maintaining all configurations identical except for the $use\_llm\_embedding$ parameter. Historical data windows are set to 4 weeks/6 days/1 hour, providing retrieval candidates for the LLM. Model capacity is reduced ($d\_model = 8$) to accommodate small-scale networks and avoid overfitting, while intervention frequency is increased to 2 to enhance causal learning effectiveness in small networks. As shown in Table 11, this control ensures that performance differences can be attributed to the LLM retrieval mechanism.

Table 11: LLM comparison experiment configuration (5-node subset)

| Parameter | Configuration Value |
|---|---|
| num_of_vertices | 5 |
| num_weeks/days/hours | 4/6/1 |
| d_model | 8 |
| d_ff_* | 16 |
| n_belt_block | 1 |
| intervention_frequency | 2 |
| num_interventions | 5 |
| batch_size | 128 |
| learning_rate | 0.0005 |
| **Unique Variable** | |
| use_llm_embedding | False $\to$ True |

### C.1.3 ABLATION EXPERIMENT CONFIGURATION

Ablation experiments are based on the complete PEMS08 configuration, with different components controlled through the ablation_mode parameter. As shown in Table 10 (main text), we systematically evaluate 8 key components, with all ablation variants maintaining identical training configurations to ensure fair comparison.

### C.1.4 DREAM3 CAUSAL DISCOVERY CONFIGURATION

To quantitatively evaluate Orion's causal discovery capability, we conduct experiments on the DREAM-3 gene regulatory network dataset. Unlike the main application scenario (traffic forecasting), DREAM-3 is configured as a static causal discovery task: we set the number of temporal

segments $S = 1$ to learn a single time-invariant regulatory network $C \in \mathbb{R}^{100 \times 100}$, adopting single-step lag modeling (gene i at time t influences gene j at time $t + 1$), which aligns with the standard setting of the DREAM challenge.

Due to the specificity of the DREAM-3 dataset (only 21 time points), specialized configurations are required to prevent overfitting: (1) a small learning rate (1e-5) and lightweight model (d_model=8) are used to adapt to the small data scale (21 time points × 100 samples); (2) temporal segments are set to 1 since gene expression data lacks daily periodicity; (3) the causal threshold is set to 0.05, stricter than traffic data (0.1), to avoid false positive edges in biological networks. These configurations (Table 12) are specifically designed for the unique characteristics of biological time series data.

The do-intervention validation mechanism is still employed in this scenario (K=25 genes randomly intervened, with effects propagated through GRU). The training process does not use the ground truth network, and evaluation metrics such as AUC are calculated only during assessment using the official DREAM scripts.

### C.1.5 INTERVENTION IMPLEMENTATION DETAILS

**Intervention node selection:** At the beginning of each batch, we use *torch.randperm(N)[:K]* to randomly select K nodes (K=30 for traffic data, K=25 for epidemic/DREAM-3 data). The selection process is independent of the current model state and time period.

**Intervention value generation:** For the selected node i, the intervention value is generated through $v_i = \tanh(g_i(c_i))$, where $g_i$ is a node-specific learnable function and $c_i$ is noise sampled from a standard normal distribution.

**Intervention implementation:** During GRU propagation, the input of the intervened node is forced to be $v_i$, and its incoming edges are truncated, simulating the semantics of the do-operator.

**Validity calculation:** The intervention effect $\Delta \hat{y}_j$ is calculated by comparing predictions before and after intervention, then compared with $C_{ij}$ to obtain the validity score (Equation 5). The entire process is differentiable and trained end-to-end.

### C.2 K-MEANS NODE SELECTION STRATEGY

Due to the computational complexity of LLM retrieval being $O(N \times H)$, where $N$ is the number of nodes and $H$ is the number of historical candidates, direct application on large-scale networks would incur prohibitive computational overhead. Therefore, we employ K-means clustering to select representative node subsets for LLM comparison experiments.

The node selection process comprises four steps:

**Step 1: Feature Extraction.** For each node $i$, we extract a four-dimensional feature vector from its time series data:

$$\mathbf{f}_i = [\mu_i, \sigma_i, \rho_i(12), r_i^{peak}] \quad (20)$$

where $\mu_i$ and $\sigma_i$ are the mean and standard deviation of node $i$'s traffic flow respectively, $\rho_i(12)$ is the autocorrelation coefficient at lag 12 steps (1 hour) capturing short-term periodicity, and $r_i^{peak}$ is the peak hour flow ratio reflecting the node's functional characteristics.

Table 12: Dream3 causal discovery experiment configuration

| Parameter | Configuration Value |
|-----------|---------------------|
| num_of_vertices | 100 |
| learning_rate | 1e-5 |
| d_model | 8 |
| num_time_segments | 1 |
| batch_size | 16 |
| causal_threshold | 0.05 |

**Step 2: K-means Clustering.** We execute standard K-means algorithm on the feature matrix $\mathbf{F} = [\mathbf{f}_1, ..., \mathbf{f}_N]^T$, setting cluster number $k = 5$ to obtain cluster centers $\mathbf{C} = \{\mathbf{c}_1, ..., \mathbf{c}_5\}$.

**Step 3: Representative Node Selection.** From each cluster, we select the node closest to the centroid as representative:

$$n_j^* = \arg\min_{n \in S_j} ||\mathbf{f}_n - \mathbf{c}_j||_2 \quad (21)$$

where $S_j$ denotes the node set of the $j$-th cluster.

**Step 4: Connectivity Verification.** We examine the connectivity of selected nodes in the adjacency matrix to ensure the subnetwork maintains certain spatial connectivity.

Table 13: K-means node selection results for PEMS08 and METR-LA datasets

| New ID | PEMS08 | | METR-LA | |
|---|---|---|---|---|
| | Original ID | Connectivity | Original ID | Connectivity |
| 0 | 9 | Connected | 29 | Isolated |
| 1 | 10 | Connected | 67 | Isolated |
| 2 | 25 | Connected | 71 | Connected |
| 3 | 31 | Connected | 127 | Isolated |
| 4 | 166 | Connected | 156 | Connected |

As shown in Table 13, the METR-LA subset contains 3 isolated nodes (nodes 29, 67, 127 with no edge connections), reflecting K-means' characteristic of selecting based on temporal features rather than spatial topology. This selection strategy ensures diversity in temporal patterns, providing comprehensive test scenarios for LLM-enhanced retrieval—encompassing both spatially correlated node pairs and spatially independent nodes with similar temporal patterns, thereby thoroughly evaluating LLM retrieval effectiveness across different scenarios.

## C.3 THREE-STAGE TRAINING WEIGHT CONFIGURATION

The three-stage training strategy employs carefully calibrated loss component weights to progressively transition from feature learning to causal discovery. Table 14 details the individual causal loss component weights during Stage 2 , where $\mathcal{L}_{causal\_total} = w_{sparse}\mathcal{L}_{sparse} + w_{valid}\mathcal{L}_{validity} + w_{conn}\mathcal{L}_{connect} + w_{temp}\mathcal{L}_{temporal} + w_{dag}\mathcal{L}_{dag}$. Table 15 presents the complete loss weight evolution across all three stages.

Table 14: Causal loss component weights in Stage 2.

| Causal Loss Component | Weight ($w_i$) |
|---|---|
| Sparsity ($\mathcal{L}_{sparse}$) | 0.01-0.02* |
| Validity ($\mathcal{L}_{validity}$) | 0.5 |
| Connectivity ($\mathcal{L}_{connect}$) | 0.1 |
| Temporal ($\mathcal{L}_{temporal}$) | 0.01 |
| DAG ($\mathcal{L}_{dag}$) | 0.1 |

Table 15: Loss weight distribution across training stages

| Loss Component | Stage 1 | Stage 2 | Stage 3 |
|---|---|---|---|
| Prediction ($\mathcal{L}_{pred}$) | 1.0 | 1.0 | 1.0 |
| Sparsity | 0 | 0.0005 | 0.0003 |
| Validity | 0 | 0.025 | 0.009 |
| Connectivity | 0 | 0.005 | 0.0015 |
| Temporal | 0 | 0.0005 | 0.00015 |
| DAG | 0 | 0.005 | 0.0015 |
| Weight Decay | 0.0001 | 0.0001 | 0.0001 |

The sparsity weight (*) is dynamically adjusted based on validity scores: when average validity exceeds 0.5, the weight decreases to 0.01 to encourage more connections; otherwise, it increases to 0.02 to enforce sparsity. This adaptive mechanism balances between discovering sufficient causal relationships and avoiding spurious connections.

## C.4 MULTI-SCALE TEMPORAL CONVOLUTION ABLATION

We conducted systematic ablation experiments on the selection of kernel numbers in multi-scale temporal convolution to explain the question of "why we choose 8 kernels".

**Experimental Setup**: All experiments were conducted on the PEMS-08 dataset, keeping all hyperparameters consistent except for the kernel configuration (d_model=64, n_belt_block=2, batch_size=128, learning_rate=0.005). We tested four kernel configurations: (1) Single-3: a single kernel size k=3, serving as a baseline without multi-scale modeling; (2) Sparse-4: sparse sampling

of 4 key scales [1,3,7,15], testing whether fewer kernels can cover key patterns; (3) Full-8 (our configuration): dense sampling of 8 odd scales [1,3,5,7,9,11,13,15]; (4) Dense-12: ultra-dense sampling of 12 consecutive scales [1,2,3,...,12], testing whether more kernels bring further improvements. All models were trained for 100 epochs using a three-stage training strategy on NVIDIA A100-40GB GPUs.

Table 16: Detailed Performance Comparison of Different Kernel Configurations

| Config | Horizon 3 | | | Horizon 6 | | | Horizon 12 | | | Average | | | Time (s/ep) |
| | MAE | RMSE | MAPE | MAE | RMSE | MAPE | MAE | RMSE | MAPE | MAE | RMSE | MAPE | |
|---|---|---|---|---|---|---|---|---|---|---|---|---|---|
| 3 | 9.79 | 14.62 | 7.30 | 13.24 | 20.88 | 8.41 | 17.99 | 27.81 | 10.25 | 14.33 | 22.73 | 8.71 | 52.7 |
| 4 | 8.76 | 12.08 | 7.55 | 13.35 | 20.06 | 9.62 | 17.46 | 26.51 | 11.65 | 14.21 | 21.81 | 10.16 | 57.9 |
| **8** | **9.44** | **12.98** | **8.82** | **13.13** | **19.66** | **10.41** | **15.18** | **23.37** | **11.42** | **13.29** | **20.24** | **10.55** | **52.7** |
| 12 | — | — | — | — | — | — | — | — | — | — | — | — | ∼2700 |

As shown in Table 16, the experimental results clearly demonstrate the multi-dimensional impact of kernel configuration on model performance. First, Single-3 shows significant performance degradation compared to Full-8 across all prediction horizons, particularly at long horizon (Horizon 12) where MAE increases from 15.18 to 17.99 (degradation of 18.5%) and RMSE from 23.37 to 27.81 (degradation of 19.0%), fully demonstrating that a single temporal scale cannot adequately capture the complex periodic and trend patterns in spatiotemporal data. Second, although Sparse-4 covers both the minimum (k=1) and maximum (k=15) scales, its performance still lags behind Full-8 by approximately 6.92% in MAE, indicating that intermediate scales (k=5,9,11,13) are equally critical for capturing temporal patterns between immediate response and long-term trends (such as half-hour fluctuations) and cannot be simply omitted.

Table 17: Pareto Analysis of Training Time and Performance

| Configuration | MAE | Training Time (h/100epochs) | Efficiency Ratio |
|---|---|---|---|
| Single-3 | 14.33 | 1.46 | Baseline |
| Sparse-4 | 14.21 | 1.61 | -0.08% / +10.3% |
| Full-8 | 13.29 | 1.46 | -7.26% / +0% |
| Dense-12 | — | ∼75.00 | — / +5000% |

In terms of computational efficiency, the training time of Full-8 (52.7s/epoch) is comparable to Single-3, because the efficient implementation of dilated convolutions allows the computation of multiple kernels to be parallelized; however, the training time of Dense-12 surges to approximately 45 minutes/epoch (2700s), mainly due to the larger parameter count from 12 kernels, the lack of efficiency advantages from dilated structures in consecutive small kernels (k=2,4,6...), and increased memory and computational overhead from denser feature fusion, making complete training (100 epochs) require approximately 75 hours, which is unacceptable in practical applications. Finally, we choose odd kernel sizes (1,3,5,...,15) instead of even numbers to maintain temporal symmetry, because odd kernels have a clear "current" time point at the center moment, while even kernels introduce a half-step temporal offset, which may cause time alignment issues in causal inference.

Full-8 is clearly superior in Pareto efficiency: compared to Single-3, it achieves 7.26% MAE improvement without increasing training time; compared to Sparse-4, it achieves 6.48% further improvement with even shorter training time (benefiting from better parallelization), while the training cost of Dense-12 makes it impractical in real applications.

Based on systematic ablation experiments, the 8 odd kernel sizes [1,3,5,7,9,11,13,15] achieve the optimal balance among performance, computational efficiency, and interpretability, and are our recommended configuration for PEMS-08 and all other datasets.

## D  LLM Fine-tuning Details

Our LLM is specifically fine-tuned on spatiotemporal data using a dataset of $\sim$100,000 time series–prediction pairs from PEMS03. Each sample contains 240 historical steps (one month of 5-minute data) with 12-step predictions. This window design matches our tri-period architecture: hourly (recent hours), daily (past 7 days), and weekly (past 4 weeks). Similarity is computed with weights $W_{cosine} = 0.7$ and $W_{euclidean} = 0.3$ to balance semantic matching.

### D.1  Dataset Construction and Preprocessing

The construction of the fine-tuning dataset is critical for enabling the LLM to understand time series prediction semantics. We systematically extracted approximately 100,000 training samples from raw traffic flow data, with each sample containing 240 historical observations and their corresponding 12-step prediction targets. The data is organized in ChatML format with the following structure:

```
{
  "messages": [
    {
      "role": "user",
      "content": "[240-step historical sequence],
                  Predict the traffic flow in the next 12 time steps."
    },
    {
      "role": "assistant",
      "content": "[12-step future predictions]"
    }
  ]
}
```

This conversational format enables the model to naturally understand the task semantics of "given historical sequence, predict future values." During data preprocessing, all values are normalized to the $[0, 1]$ interval and retain 4 decimal places to balance information preservation with token efficiency. Considering the importance of recent data in time series prediction and the input token length limitations of the original LLM, we employ a backward truncation strategy: when the sequence length exceeds max_length=2048, we prioritize retaining the most recent observations. This design choice is motivated by two factors: (1) recent data typically carries more predictive information in time series forecasting; (2) early patterns repeatedly appear in the training set, allowing the model to learn these patterns effectively.

### D.2  Training Configuration and Optimization Strategy

The fine-tuning process employs a parameter-efficient training strategy. The specific configurations are shown in Table 18:

Table 18: Fine-tuning hyperparameters and configurations

| Hyperparameter | Value | Description |
|---|---|---|
| Batch size | 16 | Effective batch per GPU |
| Gradient accumulation | 2 | Equivalent batch size of 32 |
| Initial learning rate | 2e-5 | AdamW optimizer learning rate |
| Training epochs | 5 | Prevent overfitting |
| Warmup steps | 200 | Linear warmup strategy |
| Max sequence length | 2048 | Context window limit |
| Weight decay | 0.01 | L2 regularization coefficient |

Training employs gradient checkpointing (Chen et al., 2016) to reduce memory usage while utilizing mixed precision (Micikevicius et al., 2017) training to improve efficiency. The learning rate schedule adopts a linear warmup followed by constant strategy to ensure stable model convergence.

## D.3 Training Process and Convergence Analysis

During the training process, model perplexity steadily decreased from an initial 8.3 to 2.1, indicating successful learning of the semantic representation of spatiotemporal data. The validation loss curve shows the model stabilized after epoch 3 without overfitting phenomena. This is attributed to our early stopping strategy and moderate regularization settings.

The fine-tuned model demonstrates powerful recognition capabilities for temporal patterns. Even when the absolute value ranges of input sequences differ, the model can accurately identify typical traffic patterns such as "morning peak-off-peak-evening peak" and retrieve the most similar historical patterns for matching. For instance, the model successfully recognizes that a sequence with pattern "early peak - stable flow - evening peak" semantically matches similar daily patterns despite occurring on different calendar days. This semantic understanding capability forms the core foundation of our intelligent retrieval mechanism.

# E Datasets and Evaluation

## E.1 Dataset Statistics

Table 19: Dataset statistics

| Dataset | Steps | Nodes | Features | Feature Names | Interval | Time Span | Value Range | Unit |
|---------|-------|-------|----------|---------------|----------|-----------|-------------|------|
| PEMS04 | 16,992 | 307 | 3 | Flow
Occupancy
Speed | 5min | Jan-Feb 2018 | [0, 919]
[0, 0.772]
[3, 85.2] | veh/h
ratio
mph |
| PEMS08 | 17,856 | 170 | 3 | Flow
Occupancy
Speed | 5min | Jul-Aug 2016 | [0, 1147]
[0, 0.896]
[3, 82.3] | veh/h
ratio
mph |
| METR-LA | 34,272 | 207 | 1 | Speed | 5min | Mar-Jun 2012 | [0, 70] | mph |
| CA-COVID | 335 | 55 | 3 | Cases
Population
Change Rate | Daily | Feb-Dec 2020 | [0, 7928]
[0, 7602]
[-2263, 2446] | count
normalized
count/day |
| TX-COVID | 335 | 251 | 3 | Cases
Population
Change Rate | Daily | Feb-Dec 2020 | [0, 2149]
[0, 2106]
[-569, 381] | count
normalized
count/day |

Table 19 presents comprehensive dataset statistics. The traffic datasets exhibit high-frequency sampling with rich feature dimensions, while epidemic datasets present unique challenges with daily granularity and smaller sample sizes. This diversity enables thorough evaluation of Orion's generalization capabilities across varying temporal scales and domain characteristics.

## E.2 Evaluation Metrics Definition

We employ three standard evaluation metrics to comprehensively assess prediction performance:

$$\text{MAE} = \frac{1}{N \times H} \sum_{i=1}^{N} \sum_{t=1}^{H} |y_i^{(t)} - \hat{y}_i^{(t)}| \tag{22}$$

$$\text{RMSE} = \sqrt{\frac{1}{N \times H} \sum_{i=1}^{N} \sum_{t=1}^{H} (y_i^{(t)} - \hat{y}_i^{(t)})^2} \tag{23}$$

$$\text{MAPE} = \frac{100\%}{N \times H} \sum_{i=1}^{N} \sum_{t=1}^{H} \left| \frac{y_i^{(t)} - \hat{y}_i^{(t)}}{\max(|y_i^{(t)}|, \epsilon_{\text{thresh}})} \right| \tag{24}$$

where $y_i^{(t)}$ and $\hat{y}_i^{(t)}$ represent the ground truth and prediction for node $i$ at time $t$ respectively, $N$ is the number of nodes, $H$ is the prediction horizon, and $\epsilon_{\text{thresh}} = 5$ is a threshold to handle near-zero values in traffic datasets.

MAE measures average absolute deviation, providing a linear penalty for errors and being robust to outliers. RMSE applies quadratic penalty to errors, making it more sensitive to large deviations and suitable for applications where large errors are particularly undesirable. MAPE provides a scale-independent percentage error measure, enabling fair comparison across different value ranges, though it requires careful handling of near-zero values.

### E.3 DATA PREPROCESSING

Traffic datasets undergo preprocessing including IQR-based outlier detection (Tukey et al., 1977) and Savitzky-Golay smoothing (Steinier et al., 1972) to handle sensor noise, while epidemic datasets retain raw values due to their daily aggregation nature. The preprocessing pipeline consists of: (1) missing value imputation using linear interpolation for gaps less than 3 time steps and forward-filling for longer gaps; (2) outlier detection using the interquartile range method with a factor of 1.5; (3) noise reduction through Savitzky-Golay filtering with window length 7 and polynomial order 2 for traffic data only; (4) normalization using training set statistics for all splits to prevent data leakage.

## F SUPPLEMENTARY EXPERIMENTAL RESULTS

### F.1 PERFORMANCE COMPARISON VISUALIZATION

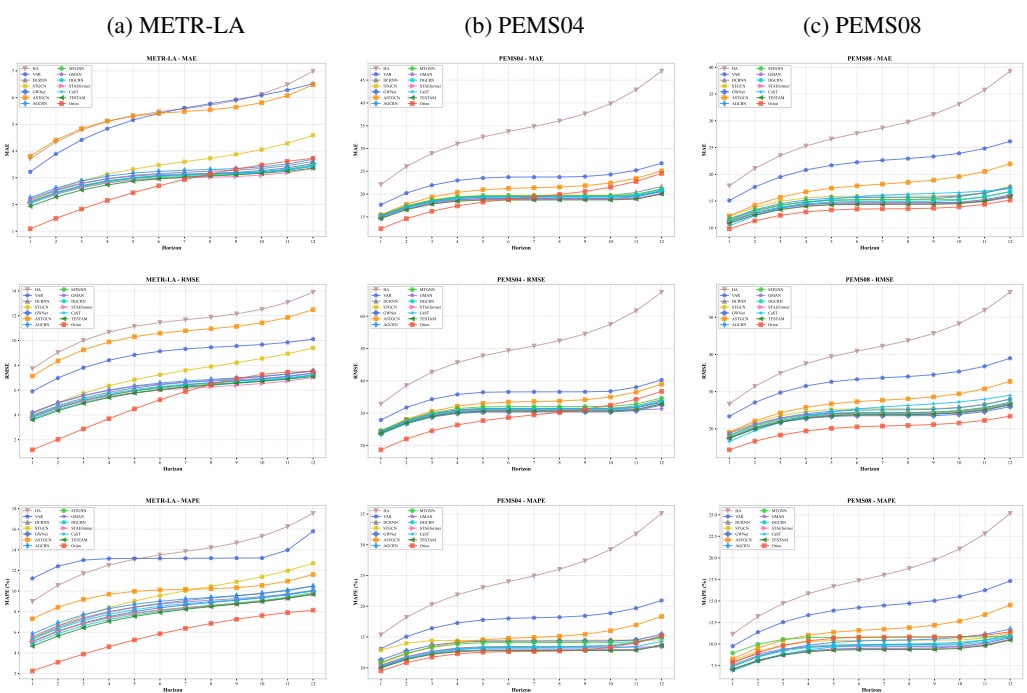

(a) METR-LA          (b) PEMS04          (c) PEMS08

Figure 8: Performance comparison across 12 prediction horizons on three traffic datasets.

Figure 8 visualizes the complete 12-horizon performance trajectories of all methods across three traffic datasets, revealing distinct performance clustering. Taking METR-LA as an example, statistical methods (HA, VAR) form the highest error cluster with steep degradation curves. Deep learning methods converge to a competitive band between MAE 3.0-4.0, with marginal differences reflecting architectural variations. Orion establishes a clearly separated lower trajectory, maintaining consistent advantages across all horizons. The visualization particularly highlights Orion's superior early-horizon performance, where causal relationships exhibit strongest predictive power before uncertainty accumulation affects long-term predictions.

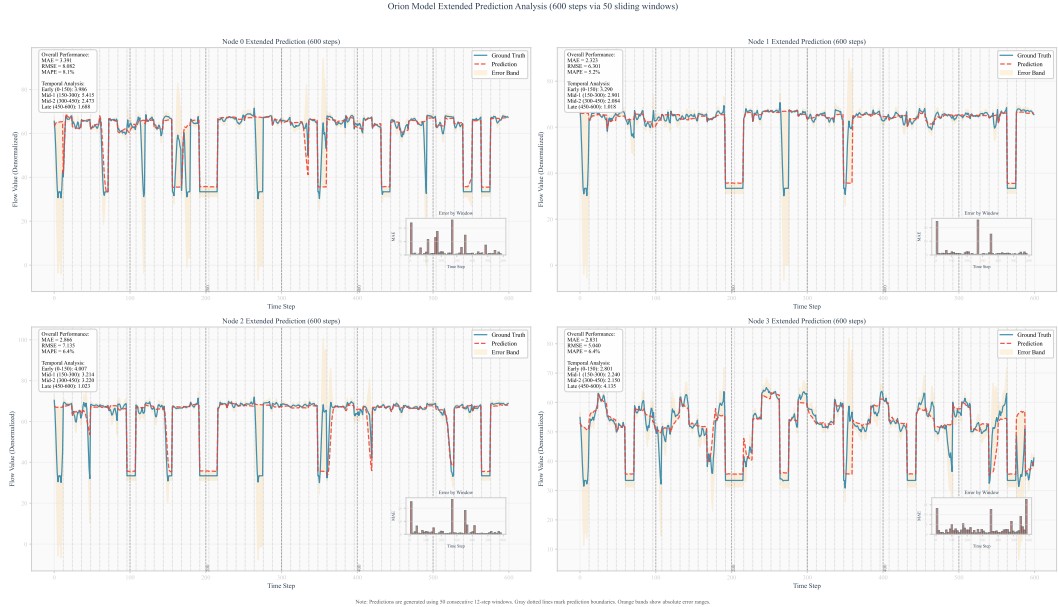

Figure 9: Extended prediction analysis for 4 representative nodes over 600 time steps through 50 consecutive 12-step sliding windows. Blue lines show ground truth, red dashed lines show predictions, orange bands indicate error ranges, and gray dotted lines mark prediction window boundaries.

## F.2 METR-LA SUPPLEMENTARY ANALYSIS

**Extended Prediction Results Analysis.** Figure 9 demonstrates Orion's 600-step prediction results on 4 representative nodes from the METR-LA dataset, with red prediction curves showing high concordance with blue ground truth values. Node 0 displays distinct V-shaped valleys with flow dropping from 60 to 20, where Orion almost perfectly reproduces these abrupt changes. Node 1's smoother pattern achieves the best MAE of 2.323 among all nodes. Node 2 exhibits sudden changes—flow plummeting from 80 to 20 or surging from 40 to 100—with Orion accurately capturing both timing and magnitude, keeping prediction error bands within reasonable bounds. This sensitivity stems from the multi-scale temporal convolution module with dilated convolutions (Yu & Koltun, 2015) de-

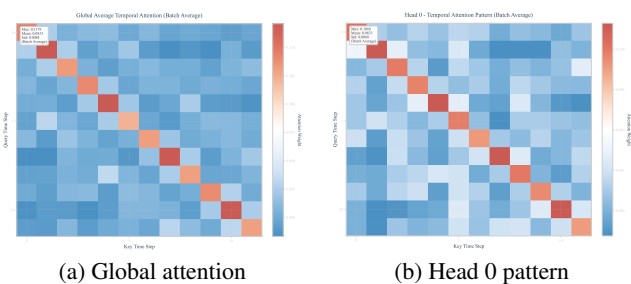

| (a) Global attention | (b) Head 0 pattern |

Figure 10: Temporal attention patterns in METR-LA. Maximum weight reaches 0.1179 with standard deviation of only 0.0084, demonstrating high stability.

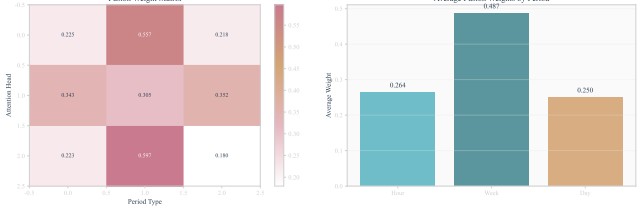

Figure 11: Multi-period fusion weight analysis.

tecting both short-term disruptions and long-term trends. Node 3 presents unique oscillations between 40-70, combining regular periodicity with irregular perturbations. Orion avoids both oversmoothing and noise-induced spurious peaks, maintaining 6.4% MAPE even in complex segments. At extreme valleys approaching flow of 10, the model shows slight overestimation but quickly self-corrects, demonstrating the generalization capability of TE-CausGAT's learned causal relationships.

**Temporal Attention Weight Analysis.** Temporal attention weight analysis (Figure 10) reveals the model's sophisticated temporal modeling mechanism. The global average attention heatmap displays clear diagonal dominance with maximum weight reaching 0.1179, concentrated on adjacent

time steps, embodying traffic flow's Markovian properties. More interestingly, secondary attention peaks appear at t-6 and t-8 positions, corresponding exactly to historical data from 1-1.5 hours ago (12 steps × 5 minutes). This "bimodal" attention pattern enables the model to simultaneously capture immediate dynamics and hourly periodicity, perfectly matching urban traffic temporal characteristics.

**Multi-Period Fusion Weight Analysis.** Multi-period fusion weight analysis (Figure 11) reveals surprising yet reasonable patterns. Weekly period data receives the highest weight of 48.7%, far exceeding hourly (26.4%) and daily (25.0%) periods. This finding highly aligns with METR-LA's actual traffic characteristics—Los Angeles traffic shows significant differences between weekdays and weekends, with weekly patterns' importance surpassing daily variations. This adaptive weight allocation mechanism enables Orion to automatically adjust multi-scale information fusion strategies based on different datasets' characteristics.

### F.3    PEMS DATASETS SUPPLEMENTARY ANALYSIS

Since the PEMS04 and PEMS08 datasets lack precise geographic coordinates for sensors, we cannot perform spatial causality visualization analysis similar to METR-LA. This section focuses on detailed visualizations of prediction performance and attention mechanisms on these two datasets.

#### F.3.1    EXTENDED PREDICTION PERFORMANCE

Figures 12 and 13 present prediction results on the PEMS datasets, which exhibit different traffic characteristics and challenges.

**PEMS08 Regular Pattern Recognition.** PEMS08 displays highly regular traffic patterns. Nodes 0 and 1 exhibit clear rectangular wave patterns—rapidly switching between high-flow plateaus and low-flow valleys. Orion precisely identifies these binary state transitions, with the red prediction line accurately responding at each transition point without delay or overshoot. Particularly for Node 0's multiple deep valleys (flow dropping to near zero), the model not only predicts the extreme low values at valley bottoms but also accurately captures their duration. Node 2 demonstrates more gradual fluctuations, where Orion similarly achieves good fitting with only slight underestimation at individual peaks.

**PEMS04 Sparse Flow Challenges.** The most distinctive feature of the PEMS04 dataset is extreme flow sparsity—multiple nodes maintain near-zero flow for prolonged periods. Nodes 0 and 2 frequently exhibit complete flow cessation, posing severe challenges for prediction models. Orion demonstrates remarkable adaptability: accurately maintaining zero predictions during zero-flow periods and responding promptly when flow resumes. However, Node 3's anomalous MAPE value exposes limitations of the evaluation metric—when ground truth values approach zero, even minimal absolute errors can cause enormous relative errors. Nevertheless, visually, the red prediction line still closely adheres to the blue ground truth, indicating the model's actual performance far exceeds what the MAPE metric suggests.

#### F.3.2    TEMPORAL ATTENTION PATTERNS

Figures 14 and 15 illustrate the temporal attention weight distributions for both datasets. PEMS08's global average attention presents a clear diagonal-dominant pattern with maximum weight of 0.0882 concentrated on adjacent time steps and a standard deviation of only 0.0023, reflecting high attention stability. Head 0's attention pattern reveals richer structure with a secondary peak at position t-6, corresponding to historical data from 1 hour ago (12 steps × 5 minutes), capturing traffic flow's hourly periodicity.

PEMS04's attention distribution is more concentrated with maximum weight of 0.0864 and standard deviation reduced to 0.0010. This tighter attention distribution may relate to PEMS04's shorter time span (2 months), with the model tending to rely on more recent observations. The reinforced diagonal pattern indicates that traffic flow in this dataset exhibits stronger Markovian properties.

Notably, fusion weights for both datasets show consistency across attention heads, with minimal weight distribution differences between different heads, validating the stability and robustness of our multi-head fusion mechanism.

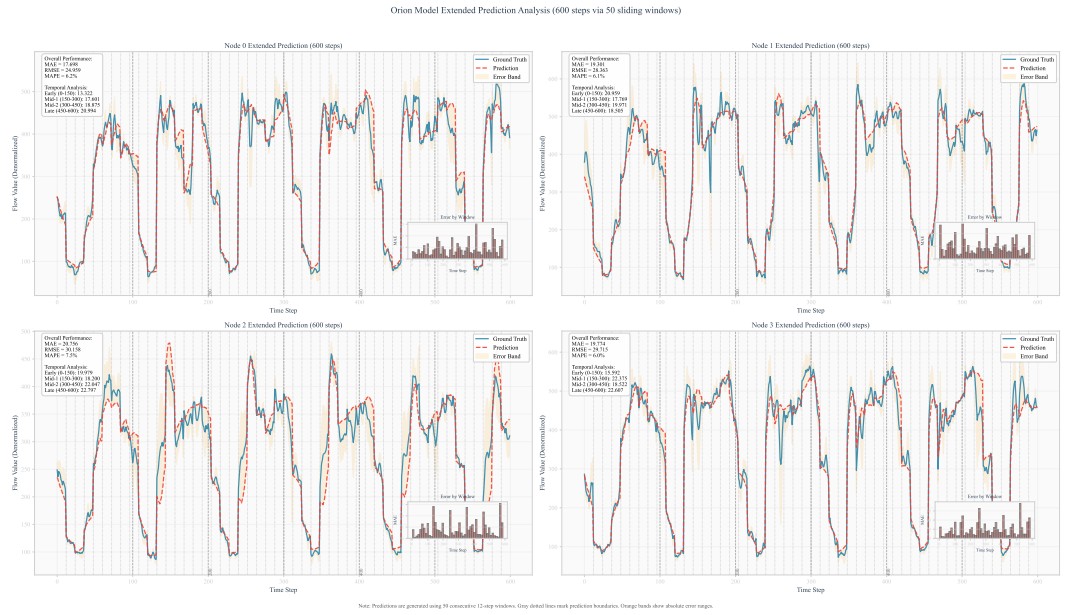

Figure 12: Extended prediction results on PEMS08 dataset for nodes 0-3. Predictions are generated using 50 consecutive 12-step sliding windows for 600 time steps. Gray dotted lines mark prediction boundaries and orange bands show absolute error ranges.

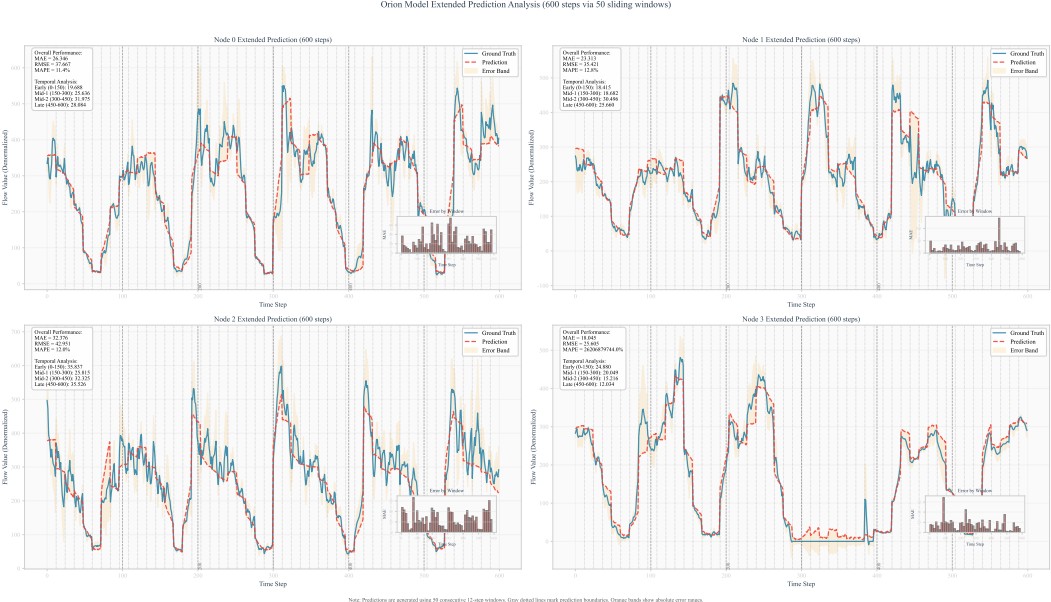

Figure 13: Extended prediction results on PEMS04 dataset for nodes 0-3.

### F.3.3 MULTI-PERIOD FUSION WEIGHTS

The weight distribution of the multi-period fusion mechanism provides insights into the model's temporal scale preferences. As shown in Figure 16, on the PEMS08 dataset, the average fusion weights for hour, week, and day periods are 0.337, 0.322, and 0.341 respectively, showing relatively balanced distribution. The batch-averaged standard deviation of the fusion weight matrix is 0.015, indicating certain dynamics in weight allocation across different time segments.

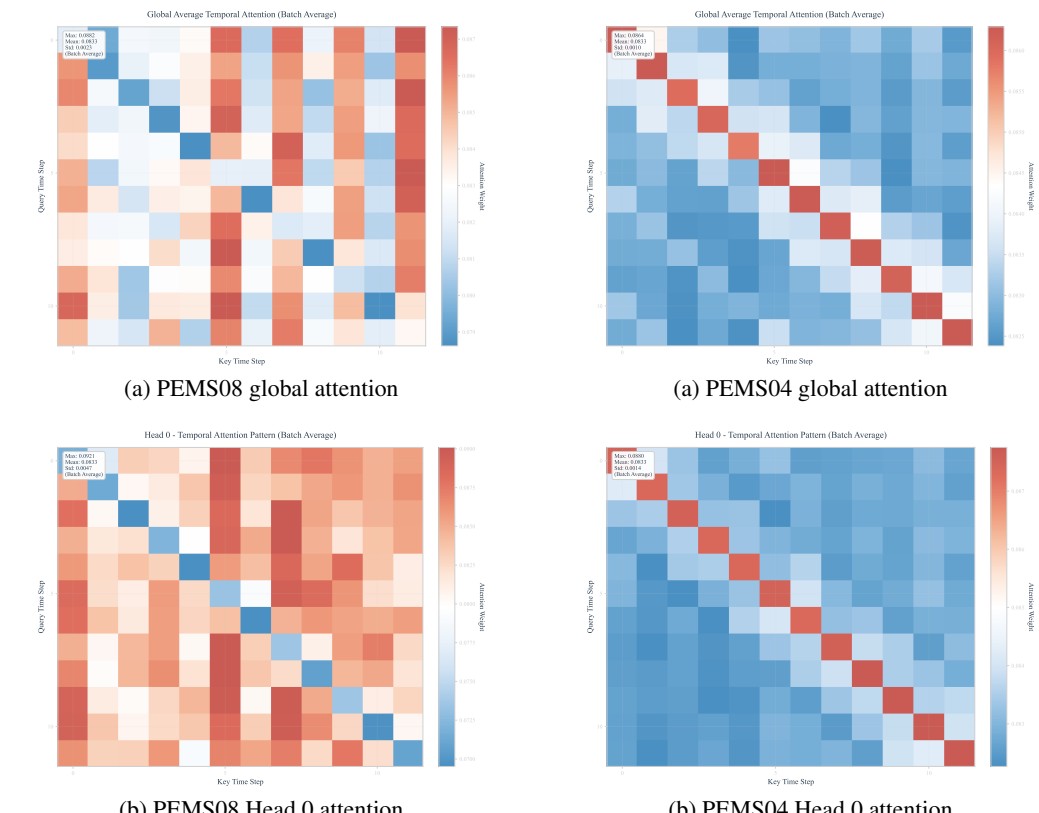

(a) PEMS08 global attention  (a) PEMS04 global attention

(b) PEMS08 Head 0 attention  (b) PEMS04 Head 0 attention

Figure 14: PEMS08 temporal attention patterns showing diagonal dominance.

Figure 15: PEMS04 temporal attention patterns with concentrated diagonal structure.

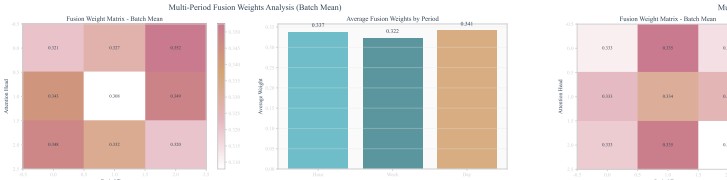

Figure 16: PEMS08 multi-period fusion weight distribution and analysis.

Figure 17: PEMS04 multi-period fusion weight distribution and analysis.

Figure 17 shows that PEMS04 exhibits more uniform weight allocation, with all three periods extremely close to 0.333 and a standard deviation of only 0.001. This highly balanced allocation pattern suggests that on the PEMS04 dataset, the model considers information from all three temporal scales equally important. This may reflect stronger regularity in traffic patterns within this dataset, where periodic features at different scales all provide valuable predictive information.

# G  LARGE LANGUAGE MODEL USAGE STATEMENT

In accordance with ICLR 2026 requirements, we disclose the auxiliary use of large language models (LLMs). LLMs were used only for non-core tasks such as LaTeX formatting and English expression polishing. All research ideas, experimental designs, analyses, and conclusions were completed solely by the authors; LLMs did not participate in any research conception or experimentation. The authors take full responsibility for the paper's content, and the use of LLMs is strictly auxiliary and does not constitute a substantive research contribution.

