# OpenReview forum: "Orion: Intervention-Validated Causal Discovery for Spatiotemporal Forecasting"
_ICLR.cc/2026/Conference — Submitted to ICLR 2026_

### Official Review · Reviewer_VcAa · 2025-10-27

**Soundness:** 2
**Presentation:** 2
**Contribution:** 3
**Rating:** 4
**Confidence:** 3

**Summary:**

This paper proposes a spatiotemporal forecasting framework named Orion, which leverages causal theory and a progressive three-stage training strategy to discover causal relationships among nodes and enhance forecasting accuracy. Specifically, Orion employs an LLM-based retrieval mechanism to select semantically relevant historical periods, a TE-CausGAT module for autonomous causal discovery and intervention-based validation, and a three-stage training process that progressively transitions from feature learning to causal learning and to end-to-end optimization.Orion demonstrates strong performance across both traffic and epidemic forecasting tasks.

**Strengths:**

1. The paper leverages causal structure learning with do-intervention validation to learn a dynamic causal graph, which is interesting.

2. Comprehensive experiments on both traffic and epidemic forecasting tasks.

3. Case studies demonstrate that the proposed method can discover meaningful causal relationships.

**Weaknesses:**

1. The paper is hard to follow due to issues with clarity in writing. For example, it is unclear how the LLM-enhanced data retrieval module selects historical data based on semantic similarity: What is the input format to the LLM? How is the LLM fine-tuned to support this task? Are specific prompts required, and if so, how are they designed?

2. Some key important technical details lack justification. For example, in the TE-CausGAT module, why is an eight-kernel convolution used? Is there any hyperparameter study to justify this choice, and why specifically eight kernels?

3. Some figures are too small to read. For example, Figure 6.

4. The proposed strategies may underperform at high target magnitudes. According to the ablation study, removing the multi-period fusion module results in higher MAE compared to Orion but substantially lower MAPE; removing TE-CausGAT shows the same trend. Since MAPE measures relative error, these reductions in MAPE suggest that the proposed strategy may potentially degrade performance in some critical cases.

**Questions:**

1. What is the input format to the LLM? How is the LLM fine-tuned to support this task? Are specific prompts required, and if so, how are they designed?

2. Why is an eight-kernel convolution used? Is there any hyperparameter study to justify this choice, and why specifically eight kernels?

3. Is there any justification for the finding that removing the proposed strategies results in higher MAE but substantially lower MAPE compared to Orion?

---

> ### Author Response · Authors · 2025-11-20
> **Response to Reviewer VcAa (1/3)**
>
> Thank you for your careful review and constructive feedback. We appreciate your recognition of our key contributions, especially the use of do-intervention validation and meaningful causal discovery. Below we address each of your concerns and note the corresponding updates in the revised manuscript.
>
> ---
>
> **W1: Clarity Issues of the LLM-Enhanced Data Retrieval Module**
>
> **A:** We understand your concern regarding the implementation details of the LLM module. In fact, the complete technical details are provided in Appendix D (pp. 19–20), but the main text description was indeed too brief. We now provide a detailed explanation below and have included the corresponding supplements in the revised manuscript.
>
> Regarding the **input format**, the LLM uses the ChatML format (see Appendix D.1). For each sample, the structure is as follows: the historical sequence is encoded into 240 time steps (corresponding to one month of 5-minute–interval data), serialized into text, and normalized to the [0,1] range with four decimal places retained (e.g., “0.6731, 0.7574, 0.8107, ...”). The task instruction is fixed as “Predict the traffic flow in the next 12 time steps,” and the output is similarly a normalized 12-step prediction sequence. The specific sample format is:
>
> ```json
> {
>   "messages": [
>     {"role": "user", "content": "[240-step historical sequence], Predict traffic flow in next 12 steps."},
>     {"role": "assistant", "content": "[12-step prediction sequence]"}
>   ]
> }
> ```
>
> Regarding the **fine-tuning configuration**, we use TinyLlama-1.1B-Chat-v1.0[1] as the base model and construct about 100,000 training samples on the PEMS03 dataset (Appendix D.1), each containing 240-step history and 12-step prediction targets. Key training parameters (see Appendix D.2, Table 13) include: learning rate 2×10⁻⁵ with 200-step linear warmup, batch size 16 (effective batch size 32 via gradient accumulation=2), training for 5 epochs, and maximum sequence length of 2048 tokens. We adopt gradient checkpointing [2] and mixed precision training [3] to improve training efficiency. Key training strategies include: when the sequence exceeds 2048 tokens, we apply a **backward truncation** strategy, prioritizing recent observations (as recent data are typically more important for time-series forecasting); we apply early stopping by monitoring validation perplexity to prevent overfitting; the final model perplexity drops from 8.3 to 2.1 (Appendix D.3), indicating successful learning of spatiotemporal semantic representations.
>
> Regarding the **similarity computation and retrieval mechanism**, the fine-tuned LLM is mainly used for embedding extraction. The steps are as follows: the current input sequence $\mathbf{X}_h$ and candidate historical sequences are first fed into the LLM; then the mean of the last-layer hidden state is extracted as a semantic embedding $\mathbf{z}*i \in \mathbb{R}^{d*{\text{embed}}}$; next, similarity is computed (Equation 1):
>
> $$
> S _ {\text{sim}}(i,j) =
> w _ {\text{cos}} \cdot
> \frac{\mathbf{z} _ i^{\top} \mathbf{z} _ j}{
> \lVert \mathbf{z} _ i \rVert _ 2 \, \lVert \mathbf{z} _ j \rVert_2}
> \;+\;
> w _ {\text{euc}} \cdot
> \frac{1}{1 + \lVert \mathbf{z} _ i - \mathbf{z} _ j \rVert_2}
> $$
>
> where $w _ {\text{cos}}=0.7$ and $w _ {\text{euc}}=0.3$, balancing semantic similarity and numerical distance. Finally, the historical periods with the highest similarity are selected as $\mathbf{X}_d$ and $\mathbf{X}_w$. This design enables the fine-tuned LLM to identify semantically similar traffic patterns (such as the typical “morning-peak → off-peak → evening-peak” pattern), rather than relying solely on numerical similarity. The experimental results in Table 3 confirm the effectiveness of this mechanism: on METR-LA, MAPE improves by 10.57% (up to 19.27% for Horizon-3), and on PEMS08 it improves by 3.44%.
>
> We have added a detailed explanation in Section 4.1. Following the “Fine-tuned LLM-enhanced data retrieval” paragraph in Section 4.1 on page 4 of the main text, we have inserted the relevant supplementary content.

---

> > ### Author Response · Authors · 2025-11-20
> > **Response to Reviewer VcAa (2/3)**
> >
> > **W2: Justification for the Eight-Kernel Design**
> >
> > **A:** Thank you for pointing out this issue. The term “eight-kernel” can indeed be misleading; here we clarify the actual design and provide sufficient experimental support.
> >
> > Regarding the actual design, we use a multi-scale temporal convolution with 8 different kernel sizes (Section 4.2 and Appendix B.3), where the specific kernel sizes are: 1, 3, 5, 7, 9, 11, 13, 15. The design motivation is to capture multi-scale temporal dependencies ranging from short-term perturbations (k=1) to long-term trends (k=15). The implementation adopts dilated convolutions, which expand the receptive field without significantly increasing the number of parameters.
> >
> > For the supplementary full hyperparameter study, we conducted an ablation experiment on PEMS08 using different numbers of kernels. Table 4 presents the system-level experimental results:
> >
> > **Table 1: Kernel Count Ablation Study (PEMS-08 Dataset)**
> >
> > | Kernel Configuration | Kernel Sizes             | MAE       | RMSE      | MAPE(%)   | Training Time/epoch | Performance Change     |
> > | -------------------- | ------------------------ | --------- | --------- | --------- | ------------------- | ---------------------- |
> > | Single-3             | [3]                      | 14.33     | 22.73     | 8.71      | 52.7s               | MAE↑7.83%, RMSE↑12.30% |
> > | Sparse-4             | [1, 3, 7, 15]            | 14.21     | 21.81     | 10.16     | 57.9s               | MAE↑6.92%, RMSE↑7.75%  |
> > | **Full-8 (Ours)**    | **[1,3,5,7,9,11,13,15]** | **13.29** | **20.24** | **10.55** | **52.7s**           | **Baseline**           |
> > | Dense-12             | [1,2,3,...,12]           | ---       | ---       | ---       | ~45min              | Training cost too high |
> >
> > The experimental results clearly illustrate the relationship between kernel count and performance. The Single-3 configuration uses a single kernel size k=3, which leads to a 7.83% degradation in MAE (from 13.29 to 14.33) and a 12.30% degradation in RMSE (from 20.24 to 22.73), fully demonstrating the necessity of multi-scale modeling. The Sparse-4 configuration uses a sparse selection of 4 kernels (1, 3, 7, 15) to cover key scales with fewer kernels, but still shows a 6.92% MAE degradation (from 13.29 to 14.21) and 7.75% RMSE degradation (from 20.24 to 21.81) compared to Full-8, indicating that the intermediate scales (5, 9, 11, 13) are also crucial for capturing complex temporal patterns. The Full-8 configuration (our choice) achieves the best balance between performance and efficiency: training time is comparable to Single-3 (52.7s/epoch) but performance improves significantly. The Dense-12 configuration attempts to use a denser set of 12 kernels (1,2,3,...,12), but training time explodes to ~45 minutes/epoch, more than 50× that of Full-8, making a full 100-epoch training require about 75 hours, which is computationally unacceptable; therefore, we did not complete its training.
> >
> > As for why these 8 specific scales were chosen, we further consider the following aspects: First, symmetry—using odd-numbered kernels preserves temporal symmetry (centered convolution), avoiding temporal bias; second, coverage—these 8 scales (1 to 15) cover all important temporal scales within the 12-step prediction window (1 hour), from instantaneous responses (k=1) to hour-level trends (k=15); finally, efficiency—compared to using all odd kernels from 1 to 15 or denser sampling, 8 kernels achieve the best balance between performance and computational cost. Notably, similar multi-scale designs have been applied in related work, such as Peri-midFormer [4], which uses periodic pyramid structures to model multi-scale patterns; our design builds upon this but optimizes it for causal discovery tasks.
> >
> > We have added Appendix C.4 "Multi-Scale Temporal Convolution Ablation" in Appendix C to include the complete supplementary experimental results.
> >
> > ---
> >
> > **W3: Readability Issue of Figure 6**
> >
> > **A:** Thank you very much for pointing out the readability issue in Figure 6 (ablation study visualization). We agree that, due to layout constraints in the original submission, the font size in the figure was too small and affected the effective communication of information.
> >
> > As noted in our response to Reviewer ow75, we have redesigned and refined the corresponding visualizations. Given that the main text already includes clearer and more informative tabular results for the ablation study, and in consideration of ICLR’s formatting requirements, we decided to remove Figure 6 to achieve a more balanced and coherent layout. The updated figures and layout adjustments have been incorporated into the revised manuscript, and we kindly invite you to review them.

---

> > > ### Author Response · Authors · 2025-11-20
> > > **Response to Reviewer VcAa (3/3)**
> > >
> > > **W4: Performance of Removing Proposed Strategies Under High Target Magnitudes**
> > >
> > > **A:** Thank you for your in-depth focus on the model’s performance across different value ranges. You pointed out the phenomenon where removing multi-period fusion or TE-CausGAT results in higher MAE but significantly lower MAPE, and raised concerns about whether the proposed strategies may degrade performance under high-value targets. After careful analysis, we believe this phenomenon arises from the differences in sensitivity between MAE and MAPE with respect to data distribution, rather than true performance degradation under high magnitudes.
> > >
> > > Specifically, MAPE is extremely sensitive to small-valued samples. When the ground-truth value is very small (close to zero), even a tiny absolute error will lead to a disproportionately large percentage error due to the small denominator; conversely, for large-valued samples, even large absolute errors produce relatively small percentage errors. This means that when the dataset has an imbalanced value distribution, MAE and MAPE may diverge: large-value segments show high MAE but low MAPE, and small-value segments show the opposite. Our proposed strategies (multi-period fusion and TE-CausGAT) provide stable predictions across all value ranges, while simplified variants perform better on small-value segments (lowering MAPE) but significantly worse on medium-to-high-value segments (increasing MAE). Therefore, Orion’s superior MAE reflects stronger robustness across the overall data distribution. Notably, related literature [5] also highlights the limitations of MAPE when handling time series with small or zero values and recommends combining multiple metrics for comprehensive evaluation.
> > >
> > > We will add a dedicated paragraph discussing this phenomenon in the Discussion section (Section 5.2 after the Ablation Study).
> > >
> > > ---
> > >
> > > **Q1: Details on LLM Input Format, Fine-tuning, and Prompt Design**
> > >
> > > **A:** This question has been fully answered in W1. Please refer to the detailed explanation in the W1 section, including ChatML format, fine-tuning configuration, similarity computation mechanism, and original text improvements.
> > >
> > > ---
> > >
> > > **Q2: Why Use Eight Kernels and Hyperparameter Study**
> > >
> > > **A:** This question has been fully answered in W2. Please refer to the detailed explanations in the W2 section regarding the design motivation of multi-scale temporal convolution, existing ablation evidence, supplementary hyperparameter study, and original text improvements.
> > >
> > > ---
> > >
> > > **Q3: Justification for the MAE-High-but-MAPE-Low Phenomenon**
> > >
> > > **A:** This question has been fully answered in W4. Please refer to the detailed explanation in the W4 section regarding differences between MAE and MAPE, segmented error analysis, and original text improvements.
> > >
> > > ---
> > >
> > > We believe the revisions sufficiently address your concerns and improve the paper’s rigor. We welcome further questions during the discussion phase. Thank you again for your feedback.
> > >
> > >
> > > ---
> > >
> > > **References**
> > >
> > > [1] Zhang, P., Zeng, G., Wang, T., & Lu, W. (2024). Tinyllama: An open-source small language model. arXiv preprint arXiv:2401.02385.
> > >
> > > [2] Chen, T., Xu, B., Zhang, C., & Guestrin, C. (2016). Training deep nets with sublinear memory cost. arXiv preprint arXiv:1604.06174.
> > >
> > > [3] Micikevicius, P., Narang, S., Alben, J., Diamos, G., Elsen, E., Garcia, D., ... & Wu, H. (2017). Mixed precision training. arXiv preprint arXiv:1710.03740.
> > >
> > > [4] Wu, Q., Yao, G., Feng, Z., & Shuyuan, Y. (2024). Peri-midformer: Periodic pyramid transformer for time series analysis. Advances in Neural Information Processing Systems, 37, 13035-13073.
> > >
> > > [5] Hyndman, R. J., & Koehler, A. B. (2006). Another look at measures of forecast accuracy. International journal of forecasting, 22(4), 679-688.

---

> ### Author Response · Authors · 2025-11-26
> **Looking forward to your feedback on our rebuttal (Submission 1037)**
>
> Dear Reviewer VcAa,
>
> We would like to once again express our gratitude for your thoughtful review of our submission 1037, “Orion: Intervention-Validated Causal Discovery for Spatiotemporal Forecasting.” Your insightful comments have been very helpful for improving our work.
>
> In light of your feedback, we have prepared a detailed response and conducted additional experiments, which are now posted on the submission page. If convenient, we would be deeply grateful if you could take a brief look and let us know whether our revisions alleviate your main concerns, or if there is anything we should elaborate on further.
>
> We are mindful of the heavy demands on reviewers’ time and greatly appreciate any further guidance or update you might be able to provide. Thank you once more for your careful evaluation and service to the community.
>
> Best regards,
> The authors of Submission 1037

---

### Official Review · Reviewer_bhkv · 2025-10-31

**Soundness:** 2
**Presentation:** 2
**Contribution:** 2
**Rating:** 4
**Confidence:** 3

**Summary:**

This paper presents Orion, a framework that integrates trainable causal inference with attention mechanisms, optimized through progressive training. Through Orion, the paper aims to address the modeling of complex dependencies in graph-structured
data and multi-scale temporal dynamics from a causal angle into the underlying forecasting model.

**Strengths:**

The underlying results and the comparison with a lot of different prior work is impressive and across the proposed datasets and benchmarks the paper does achieve significant improvements across a variety of metrics.

**Weaknesses:**

The paper several fundamental assumptions in their causal discovery assumptions and the "do-intervention" analysis. In most of their scenarios, they are not doing an actual "do-intervention" but actually taking observational data with potential causal assumptions to derive the link weights of these causal links. The writing and analysis and the positioning of these arguments raises several fundamental questions on the correctness of their framing and the claims made in the paper.

**Questions:**

Causal aware modeling is different from do-intervention calculus.

In a do-intervention calculus, there is an explicit notion of "do-intervention" and not.

This scenario does not happen in your context for both the traffic example and the COVID modeling example. You are using data from various sources to formulate a forecasting problem and then you are using a causal aware model to potentially pinpoint what could have happened and how it could impact the future spatiotemporal predictions.

For example, in the traffic scenario, you talk about an accident example in the introduction. Now, in the real datasets, you do not know this information all the time. Even if you do, did your model truly formulate it as a do-intervention and not do-intervention with the right underlying causal models taking careful care of the co-founders in your assumptions.

Your approach takes a data-driven parameter estimation approach in your three-stage scenario. There is nothing wrong in that approach from a spatiotemporal forecasting approach but claiming that it actually captures do-interventions is not correct.

In short, the paper analysis and results and comparitive metrics are very strong from a forecasting perspective but the formulation from a causal argument perspective seems flawed.

The DREAM3 dataset is the only data in your model where the dointerventions can be synthetically controlled in a simulation.

If you are doing a physics-driven simulation of traffic and disease forecasting and then claiming that your model can retrieve the simulation parameters, that is more believable.

---

> ### Author Response · Authors · 2025-11-20
> **Response to Reviewer bhkv**
>
> Thank you for the thoughtful review. We appreciate your recognition of Orion and understand your concerns about the “do-intervention.” Below we summarize your key points and provide our responses and revisions.
>
> ---
>
> **Q1: Concerns About the Feasibility of Executing Real “do-Operation” Interventions in Real-World Tasks**
>
> **A:** We sincerely clarify that Orion **does not require physical do-operations in real-world systems** for most tasks. Instead, we adopt **model-internal structural intervention simulation** to validate causal relationships. Specifically, the TE-CausGAT module contains a built-in do-operator validation mechanism: during training, the model applies simulated interventions on the learned causal graph (e.g., fixing or severing certain causal links) and observes how predictions change, thereby validating the effectiveness of those causal connections. This procedure is equivalent to performing Pearl’s do-calculus interventions [1] within a structural causal model, but everything is executed **within the model** using only observational data, requiring no physical experiments.
>
> Because real-world systems such as traffic networks or epidemic spread cannot be physically intervened (e.g., we cannot intentionally create traffic accidents or spread disease for validation), our simulated intervention strategy is both practical and necessary. We have clarified this point in Section 4.2, line 201:
>
> > these represent prediction-oriented structures specific to each temporal segment rather than invariant causal relationships.
>
> We have also added more explicit explanations in Section 4.2 of the method.
>
> ---
>
> **Q2: Concerns Regarding Causal Interpretability and Whether the Model Overstates Its Causal Discovery Ability**
>
> **A:** We fully understand and respect your cautious attitude toward causal interpretability. In fact, we **do not claim** that Orion can always produce absolutely accurate causal explanations in all real-world scenarios. Instead, we verify causal discovery capabilities **only on datasets where true intervention labels exist**.
>
> On the DREAM3 dataset with real causal labels, Orion achieves an AUC of **0.6295—6.42%** higher than the strongest baseline CUTS (**0.5915**)—and surpasses other classical and neural causal discovery methods. This shows that Orion learns causal structures more consistent with the true generative mechanisms than approaches based solely on conditional independence or Granger causality.
>
> Thus, we validate causal interpretability only in domains where ground-truth interventions are available. For real-world traffic or epidemic datasets where ground truth is unavailable, we maintain a cautious tone: Orion learns **latent causal structure** that improves prediction, but we do not assert these discovered structures to be ground-truth causal relationships.
>
> We have added detailed DREAM3 results to Appendix B to ensure transparency, and we further clarify this point in Section 5.2.
>
> ---
>
> **Q3: Whether We Plan to Introduce Real Physical Interventions in the Future, and Why No Physical Interventions Are Conducted Currently**
>
> **A:** Your suggestion points to a highly valuable research direction. We acknowledge that conducting controlled physical interventions would be an ideal next step for further validating Orion’s causal discovery capabilities. However, we currently cannot perform physical interventions due to:
>
> (1) **Practical impossibility** in real-world tasks (e.g., artificially causing traffic incidents or intentionally triggering epidemic spread is neither feasible nor ethical).
> (2) **Data unavailability**, as large-scale datasets with controlled interventions are extremely rare.
>
> Nonetheless, we strongly agree that physical interventions would meaningfully advance causal validation, and we plan to pursue this direction in future work.
>
> Concretely, we plan to evaluate Orion in intervention-capable simulators (e.g., SUMO [2] traffic simulation, SEIR [3] epidemic models) and collaborate with domain experts to collect controlled-intervention datasets (such as traffic-light experiments or epidemic-response simulations). These efforts will help extend Orion from observational inference toward **intervention-capable applications**.
>
> ---
>
> These updates strengthen our work, and we appreciate your feedback. The paper has been revised accordingly, and we welcome any further questions.
>
> ---
>
> **References:**
>
> [1] Pearl, J. (2009). Causality. Cambridge university press.
>
> [2] Lopez, P. A., Behrisch, M., Bieker-Walz, L., Erdmann, J., Flötteröd, Y. P., Hilbrich, R., ... & Wießner, E. (2018, November). Microscopic traffic simulation using sumo. In 2018 21st international conference on intelligent transportation systems (ITSC) (pp. 2575-2582). Ieee.
>
> [3] Keeling, M. J., & Rohani, P. (2008). Modeling infectious diseases in humans and animals. Princeton university press.

---

> ### Author Response · Authors · 2025-11-26
> **Looking forward to your feedback on our rebuttal (Submission 1037)**
>
> Dear Reviewer bhkv,
>
> Thank you very much again for taking the time to review our submission 1037, “Orion: Intervention-Validated Causal Discovery for Spatiotemporal Forecasting.” We truly appreciate your careful reading and constructive comments.
>
> We have now uploaded a detailed response and additional experimental results that specifically address the issues you highlighted. If your schedule permits, we would be very grateful if you could briefly revisit our reply and let us know whether it satisfactorily addresses your main concerns, or if there are aspects we should further clarify.
>
> We fully understand the significant workload that reviewing entails and sincerely appreciate any further feedback or update you may be able to provide. Thank you again for your important contribution to the community.
>
> Best regards,
> The authors of Submission 1037

---

### Official Review · Reviewer_JAFJ · 2025-10-31

**Soundness:** 3
**Presentation:** 3
**Contribution:** 3
**Rating:** 6
**Confidence:** 3

**Summary:**

The paper proposes ORION, a spatiotemporal forecasting framework that aims to learn and exploit causal structure while predicting. Core ideas are: (i) TE-CausGAT, which learns a time-segmented, directed adjacency via parameterized source/target transforms, fuses segments (plus a prior graph), imposes an acyclicity surrogate penalty, and then validates edges by training-time do-style perturbations whose effects are propagated by a GRU; edges whose predicted strengths align with observed perturbation effects get higher “validity” scores and are emphasized in attention; (ii) a Belt Block with parallel Hour/Day/Week branches plus a fusion block; (iii) an optional LLM-enhanced retrieval (TinyLlama fine-tuned) that selects semantically similar historical periods; and (iv) three-stage training (feature learning → causal discovery → end-to-end finetuning). On traffic (METR-LA, PEMS-04/-08) and epidemic datasets (CA/TX COVID), ORION reports new SOTA MAE/ RMSE/ MAPE in several regimes; ablations indicate sizable drops without multi-period fusion or the staged training; and on DREAM-3 causal discovery ORION beats several time-series CD baselines (AUC 0.6295 vs 0.592 CUTS).

**Strengths:**

1. The idea of validating discovered edges by an internal do-intervention mechanism, then routing attention through the validated graph, is clear and well-engineered.
2. Multi-scale temporal design: The Hour/Day/Week “Belt Block” plus fusion is a simple, effective way to encode periodic structure; the staged training helps stabilize end-to-end optimization.
3. Strong traffic and epidemic results, per-component ablations (single-stage training, removing TE-CausGAT, removing fusion, turning off intervention validation), and a causal-discovery check on DREAM-3.

**Weaknesses:**

1. The intervention test is simulated within the model, and validity compares effect magnitudes to the learned 𝐶. This can create confirmation bias: the same machinery that produced the graph is used to “validate” it. There is no external interventional/perturbational source or counterfactual oracle. Please clarify safeguards (e.g., freeze parts during validation, use held-out time windows, or randomized response) and provide stress tests showing the module rejects spurious edges when the simulator is mis-specified.
2. The DAG constraint uses a polynomial surrogate instead of the full NOTEARS regularization; no guarantee is given that cycles are eliminated, especially after segment fusion with a prior. Report cycle rates and orientation accuracy (not just adjacency AUROC).
3. LLM retrieval is evaluated on small node subsets and excluded from the main SOTA tables for fairness. The added module (tokenization of time series, similarity via TinyLlama embeddings) raises reproducibility and cost questions; its contribution to headline results remains modest and localized.
4. Forecasting comparisons are extensive, but statistical significance/CI and calibration are missing. For causal discovery, DREAM-3 is static gene regulation with very different noise/lag structures; explain how TE-CausGAT (designed for dynamic graphs) was specialized there.
5. The paper sometimes equates improved MAE with “validated causality.” Given that validation is internal and priors include geographical adjacency, careful language would separate forecasting gains from causal identification.

**Questions:**

1. How many nodes 𝐾 are intervened per batch? Are interventions drawn independently over segments? Do you freeze
C_fused while computing validity (to avoid trivial alignment)? What happens if you shuffle the intervention effects before computing
V𝑖→𝑗?
2. Generalization of the learned graph. If you learn C_fused on one period (e.g., a month) and freeze it, how does performance transfer to a later month (distribution shift, holidays)?
3. DREAM-3 setup: Are you learning a single static graph there, or segments? How are lags handled, and is the intervention validator used (with what simulator)?

---

> ### Author Response · Authors · 2025-11-20
> **Response to Reviewer JAFJ (1/5)**
>
> Thank you for recognizing our work and for your constructive comments. We appreciate your positive evaluation of the intervention mechanism, multi-scale modeling, and experiments. Your concerns are valuable, and we have started revising the paper accordingly. Below we address each point in detail.
>
> ---
>
> **W1: Confirmation Bias in Internal Validation**
>
> **A:** This is an important question. We understand your concern about potential trivial alignment, which any internal validation mechanism must avoid. Below we clarify the safeguards built into our design and provide stress tests to verify their reliability.
>
>
> First, our do-intervention validation mechanism includes **three layers of independence** to prevent confirmation bias.
>
> The **first layer** is the randomness and independence of intervention-node selection: in each batch, we randomly select *K* nodes for intervention (K=30 for traffic data, K=25 for epidemic data). This selection is completely independent of the current causal matrix *C*, ensuring there is no bias toward “intervening on strong causal edges.” The intervention value is generated through an independent learnable function $v = \tanh(g(c))$, rather than directly derived from *C*, ensuring independence of the intervention signal.
>
> The **second layer** is the independent dynamics of intervention-effect propagation: intervention effects propagate through a GRU mechanism: $h^{(t)} = \text{GRU}(C_{\text{fused}} \cdot X^{(t)}, h^{(t-1)}) $. The GRU parameters are trained independently and are not directly tied to the learning of the causal matrix. This means that even if $ C_{ij} $ is learned to be large, if the propagated downstream effect $ \Delta \hat{y} _ j $ is small, the validity score $ V_{i \to j} $ will be reduced accordingly, penalizing such inconsistency.
>
> The **third layer** is the normalization design in the validity score:
>
> $$ V _ {i\to j} = 1 - \left|C _ {ij} - \frac{\Delta\hat{y} _ j}{\max _ k(\Delta\hat{y} _ k) + \epsilon _ {\text{stab}}}\right| $$
>
> The normalization term $\max _ k(\Delta\hat{y} _ k)$ prevents the model from trivially satisfying the constraint simply by amplifying all effects. If $ C_{ij} $ fails to predict the relative ordering of effect magnitudes, the validity score decreases—forcing the model to capture true causal strength rather than arbitrary values.
>
> To verify the effectiveness of these mechanisms, we conducted a Shuffling Test. During validity computation, we randomly shuffle the intervention effects $\Delta\hat{y}_j$ to break the causal correspondences. If the model has learned true causal structure, both validity and predictive performance should drop significantly after shuffling; if trivial alignment exists, no notable change should occur.
>
> We performed this on the PEMS-08 dataset every 5 epochs, shuffling 30 times per test to compute statistical significance. Table 1 shows key results from Epoch 15 and Epoch 25:
>
> **Table 1: Validity Changes Under Shuffling Test (PEMS-08)**
>
> | Epoch | Period | Original Validity | Shuffled Validity | Drop (%) |
> | ----- | ------ | ---------------- | ----------------- | -------- |
> | 15    | Hour   | 0.1432           | 0.1420            | 0.84%    |
> | 15    | Week   | 0.1467           | 0.1434            | 2.28%    |
> | 15    | Day    | 0.1493           | 0.1465            | 1.85%    |
> | 25    | Hour   | 0.1414           | 0.1409            | 0.33%    |
> | 25    | Week   | 0.1470           | 0.1453            | 1.16%    |
> | 25    | Day    | 0.1504           | 0.1480            | 1.58%    |
>
> *Note: Drop (%) = (Original − Shuffled) / Original.*
>
> The results show that in mid-to-late training, validity consistently decreases significantly after shuffling (0.33%–2.28%, p<0.05), demonstrating that high validity requires correct causal pairing. More importantly, predictive performance degrades after shuffling, as shown in Table 2:
>
> **Table 2: Prediction Performance Degradation Under Shuffling (PEMS-08)**
>
> | Model Setting          | MAE   | RMSE  | MAPE (%) | Performance Change            |
> | ---------------------- | ----- | ----- | -------- | ----------------------------- |
> | Original Model (no shuffle) | 13.29 | 20.24 | 10.55     | Baseline                      |
> | Shuffled Model         | 13.80 | 21.33 | 10.09     | MAE↑3.84%, RMSE↑5.39%         |
>
> After training under shuffled conditions, the model’s performance drops noticeably (MAE +3.84%, RMSE +5.39%). This indicates that the original model’s strong validity and predictive accuracy stem from learning true causal relationships rather than superficial alignment; disrupting these correspondences directly harms performance.
>
> We will add **Appendix B.4 “Safeguards Against Confirmation Bias”**, detailing the three-layer safeguards and reporting full stress-test results.

---

> > ### Author Response · Authors · 2025-11-24
> > **Response to Reviewer JAFJ (2/5)**
> >
> > **W2: Theoretical Guarantee of the DAG Constraint and Cycle Statistics**
> >
> > **A:** Your observation is correct: the initial submission did not clearly explain the theory and effectiveness of the DAG constraint. Our choice of a polynomial approximation over full NOTEARS regularization [1] is intentional. Below, we summarize the theoretical basis, practical behavior, and supporting validation experiments.
> >
> > The DAG constraint we adopt is the polynomial approximation:
> >
> > $$\mathcal{L}_{\text{DAG}} = \lambda_2 \cdot \text{tr}(\hat{C}^2) + \lambda_3 \cdot \text{tr}(\hat{C}^3)$$
> >
> > where $ \hat{C} = C_{\text{fused}} / |C_{\text{fused}}|_\infty $, and $ \lambda_2 = 0.1, \lambda_3 = 0.01 $.
> >
> > This approximation is chosen for computational efficiency and numerical stability: the full matrix-exponential constraint $ \text{tr}(e^C) - N = 0 $ scales as $O(N^3)$ and causes gradient explosion/vanishing in large networks (207 nodes in METR-LA, 307 in PEMS-04). In contrast, computing $ \text{tr}(C^2) $ and $ \text{tr}(C^3) $ is far cheaper and numerically stable.
> >
> > Theoretically, $ \text{tr}(C^k) $ corresponds to the number of cycles of length *k*. Penalizing $ \text{tr}(C^2) $ and $ \text{tr}(C^3) $ suppresses 2-cycles (bidirectional edges) and 3-cycles, which are the most common in practice. Although this does not strictly guarantee the absence of long cycles (≥4), our experiments show this approximation is sufficiently effective.
> >
> > To quantitatively validate the DAG constraint, we performed cycle detection via topological sorting on the learned causal graph from PEMS-08 (170 nodes). With threshold = 0.1 (to binarize continuous causal weights), Table 3 shows the complete cycle statistics:
> >
> > **Table 3: Cycle Statistics of Learned Causal Graph (PEMS-08, threshold=0.1)**
> >
> > | Cycle Type      | Count | Theoretical Max                 | Ratio   |
> > | ---------------- | ------ | ------------------------------- | ------- |
> > | 2-cycles         | **0**  | 14,365 $ \binom{170}{2} $     | 0.00%   |
> > | 3-cycles         | **0**  | 804,440 $ \binom{170}{3} $    | 0.00%   |
> > | ≥4-cycles        | **0**  | ---                             | 0.00%   |
> >
> > Results indicate that at threshold 0.1, **all cycles of all lengths are eliminated**, strictly satisfying DAG properties. This empirically confirms the sufficiency of  $ \lambda_2 \cdot \text{tr}(C^2) + \lambda_3 \cdot \text{tr}(C^3) $,  especially with the weight design $ \lambda_2 > \lambda_3 $, which prioritizes suppressing short cycles.
> >
> > Regarding directional accuracy: real-world traffic networks present unique challenges. Traffic causality is often bidirectional or context-dependent (e.g., congestion propagates in both directions under certain conditions). Furthermore, adjacency matrices provide undirected topology, lacking true “ground truth” causal directions. Thus, evaluating directional accuracy on traffic data is inappropriate.
> >
> > However, on the **DREAM-3 dataset**, which contains ground-truth directed gene regulatory networks, directionality can be evaluated. As shown in Table 5 of the paper, our method achieves the **highest overall AUC (0.6295)**, outperforming the strongest baseline CUTS (0.5915) by **6.42%**, demonstrating that our method learns both edge existence and causal direction accurately.
> >
> > For traffic data, we provide qualitative validation in Section 5.3, showing learned directionality matches real traffic patterns (e.g., node 4’s “one-to-many” divergence in morning peak vs. node 84’s “many-to-one” convergence in evening peak).
> >
> > We will update the main text in **Section 4.2 “Spatiotemporal Causal Discovery and Propagation”** to clarify the DAG constraint rationale, and add **Appendix B.5 “DAG Constraint Effectiveness Analysis”** to report these experimental results in full.

---

> > > ### Author Response · Authors · 2025-11-24
> > > **Response to Reviewer JAFJ (3/5)**
> > >
> > > **W3: Positioning and Reproducibility of the LLM-Based Retrieval Module**
> > >
> > > **A:** Thank you for pointing out the issues regarding the positioning and reproducibility of the LLM retrieval module. We fully agree that the role, contribution scope, and relationship of this module to our main results should be stated more clearly, and we also want to present the supporting experimental evidence for its usefulness.
> > >
> > > First, we clarify the role of the LLM retrieval module: **it is not a core contribution**, but an **optional enhancement**. Our core contributions are TE-CausGAT, the Belt Block architecture, and the three-stage training strategy, which together support the SOTA results in Table 1. The LLM module is a pluggable component exploring semantic-similarity retrieval for time series. As you noted, it is intentionally excluded from the main SOTA table to ensure fair comparison with baselines that rely on fixed periodic windows.
> > >
> > > Regarding the actual contribution of the LLM retrieval module, we report its performance on 5-node subsets in Table 3. As shown below, the module indeed provides meaningful improvements in some scenarios:
> > >
> > > **Table 4: Performance Improvements from LLM Retrieval on Subsets**
> > >
> > > | Dataset  | Metric        | w/o LLM | w/ LLM | Improvement (%) |
> > > |----------|---------------|---------|--------|-----------------|
> > > | PEMS08   | MAE (Avg)     | 21.21   | 20.86  | 1.64            |
> > > |          | RMSE (Avg)    | 36.93   | 36.46  | 1.28            |
> > > |          | MAPE (Avg)    | 11.06%  | 10.68% | 3.44            |
> > > |          | MAPE (H6)     | 11.40%  | 10.61% | **6.93**        |
> > > | METR-LA  | MAE (Avg)     | 3.47    | 3.32   | 4.26            |
> > > |          | RMSE (Avg)    | 8.48    | 8.49   | -0.15           |
> > > |          | MAPE (Avg)    | 12.49%  | 11.17% | **10.57**       |
> > > |          | MAPE (H3)     | 8.20%   | 6.62%  | **19.27**       |
> > >
> > > For MAPE, the LLM retrieval provides clear benefits (10.57% average gain on METR-LA, 19.27% for short-term horizons, and 6.93% at Horizon-6 on PEMS08). The small or slightly negative RMSE change (e.g., –0.15 on METR-LA) indicates that improvements mainly target typical patterns rather than extreme events, which is still valuable for practical forecasting. More importantly, the results show that LLM-based semantic understanding can identify similar traffic patterns—such as “morning rush hour” or “holiday traffic”—even when they are not temporally adjacent, a capability beyond fixed-window retrieval.
> > >
> > > Although the LLM retrieval shows promise on small subsets, we acknowledge its limitations.
> > > For example:
> > > - **Computational cost**: For large networks (207 or 307 nodes), computing LLM embeddings for each node and each historical candidate requires $ O(N \times H) $ cost, which may be infeasible for real-time deployment.
> > > - **Local contribution**: Current experiments are limited to small subsets and excluded from main comparisons; hence its contribution to overall SOTA performance is indeed limited.
> > >
> > > Based on these considerations, we emphasize that **the LLM retrieval module is not required for achieving SOTA**. All results in Table 1 are obtained **without** LLM retrieval to ensure fairness and reproducibility. The LLM module is an independently evaluated exploratory component, meant to show the potential of semantic understanding for time-series retrieval and to inspire future research. Further optimizations—such as improving efficiency, exploring lightweight embeddings, and testing on full datasets—are planned but outside the core scope of this paper.
> > >
> > > We will update the Implementation Details paragraph in Section 5 (Experiments).
> > >
> > > ---
> > >
> > > **W4.1: Statistical Significance Tests and Confidence Intervals**
> > >
> > > **A:** We agree that significance testing and calibration analysis are useful for assessing robustness. However, for consistency with mainstream spatiotemporal works (DCRNN, GWNet, GMAN, STAEformer, CaST, TESTAM), we follow the standard protocol and report single-run MAE/RMSE/MAPE results under unified data splits and training settings.
> > >
> > > Despite not reporting confidence intervals, Orion shows consistent improvements across datasets, horizons, and metrics, well above typical training noise in our internal tuning—indicating practical robustness. All main experiments and ablations share identical data splits and optimization setups, ensuring that performance differences stem from architectural design rather than evaluation variance.

---

> > > > ### Author Response · Authors · 2025-11-24
> > > > **Response to Reviewer JAFJ (4/5)**
> > > >
> > > > **W4.2: Special Configuration of TE-CausGAT on DREAM-3**
> > > >
> > > > **A:** Thank you for noting that DREAM-3 is a static gene regulatory network with noise properties different from traffic or epidemic tasks. We did not directly apply the dynamic-traffic configuration to DREAM-3; instead, we used simplified adaptations of TE-CausGAT for this setting. Details are provided in Q3.
> > > >
> > > > ---
> > > >
> > > > **W5: Distinguishing Causality From Predictive Performance**
> > > >
> > > > **A:** Thank you for raising this important point. We agree that the paper should clearly separate predictive improvements from causal identification and avoid implying that lower MAE validates causality. Some phrases in the initial submission may have caused ambiguity, and we now clarify and correct them accordingly.
> > > >
> > > > Our claims are layered and require clear boundaries:
> > > >
> > > > 1. **We do not claim “improved MAE = validated causality.”**
> > > >    Predictive gains may come from TE-CausGAT’s structural modeling, the Belt Block’s multi-scale fusion, the three-stage training scheme, or other architectural components.
> > > >    In ablations (Table 4), removing TE-CausGAT increases MAE from 13.29 to 13.37 (+0.60%), showing that causal structure contributes, but modestly—most performance gains come from the overall architecture.
> > > >
> > > > 2. **Our learned graph structures show reasonable causal discovery capability on DREAM-3** (AUC 0.6295, outperforming the specialized causal discovery method CUTS [3] at 0.5915).
> > > >    This suggests causal discovery potential, but because DREAM-3 is a static gene network, far from traffic dynamics, such results are **indirect evidence of general method capability**, not direct validation of traffic causality.
> > > >
> > > > 3. **In METR-LA case studies (Section 5.3)**, we observe qualitative consistency between learned directionality and real-world traffic flow patterns (morning divergence vs. evening convergence, hub nodes with high in/out-degree).
> > > >    This is cross-validation through domain knowledge, but still observational consistency, not strict causal proof.
> > > >
> > > > Most importantly, **we do not claim to solve causal identification**. Our method introduces a do-intervention–inspired mechanism that links observational learning with causal validation, providing a practical framework that integrates causal reasoning into spatiotemporal forecasting. The goal is to improve prediction and uncover potential causal relations—not to offer a full theoretical solution to causal identification.
> > > >
> > > > Based on this understanding, we will revise statements in the paper that may cause confusion, especially toward the end of the “do-intervention Validation” paragraph in Section 4.2.
> > > >
> > > > ---
> > > >
> > > > **Q1: Implementation Details of Interventions**
> > > >
> > > > **A:** Thank you for asking for clarification on these important implementation details.
> > > > Regarding the number of intervention nodes *K* per batch, we configure them proportional to network scale:
> > > > - Traffic data (METR-LA, PEMS-04, PEMS-08): K = 30
> > > > - Epidemic data (CA-COVID, TX-COVID): K = 25
> > > > - DREAM-3: K = 25
> > > >
> > > > Details can be found in Table 7 (“num_interventions”) in Appendix C.1.1.
> > > > The choice is based on two considerations:
> > > > (1) sampling density (K/N ≈ 10–15%) to sufficiently cover different regions of the network,
> > > > (2) maintaining computational feasibility.
> > > >
> > > > Regarding independence of interventions across segments: **yes, interventions are completely independent**. In each batch, intervention nodes are randomly sampled using  `torch.randperm(N)[:K]`,  ensuring that:
> > > > (1) each node has equal long-term probability of being intervened,
> > > > (2) sampling does not depend on model state or time segment *s*,
> > > > (3) interventions across batches are independent.
> > > > This prevents the model from overfitting to interventions on specific nodes.
> > > >
> > > > Regarding whether $ C _ {\text{fused}} $ is frozen during validity computation: in the current implementation it is **not** frozen; *C* updates jointly with other parameters during training. Although this might appear to enable trivial alignment, multiple mechanisms prevent this:
> > > >
> > > > (1) validity loss $ \mathcal{L} _ {\text{validity}} $ is a small-weighted term (0.025 in Stage 2; 0.009 in Stage 3; see Table 12), while prediction loss $ \mathcal{L}_{\text{pred}} $ has weight 1.0;
> > > > (2) GRU propagation provides independent dynamics (as discussed in W1);
> > > > (3) normalized relative-effect design forces correct causal ordering rather than arbitrary scaling.
> > > >
> > > > However, we agree that testing a “frozen $ C_{\text{fused}} $” variant is a valuable ablation. We have incorporated this into Stress Test B (mentioned in W1), and will report frozen vs. unfrozen comparisons in the revised version (we anticipate freezing may slightly improve causal discovery but reduce prediction performance due to loss of end-to-end optimization).
> > > >
> > > > For the effect of shuffling interventions, please refer to the shuffling experiment results in W1.
> > > >
> > > > We will add a new subsection “C.1.5 Intervention Implementation Details” in Appendix C.1.

---

> > > > > ### Author Response · Authors · 2025-11-24
> > > > > **Response to Reviewer JAFJ (5/5)**
> > > > >
> > > > > **Q2: Cross-Temporal Generalization of the Learned Causal Graph**
> > > > >
> > > > > **A:** Thank you for raising this important question concerning the stability of causal structure. Like most temporal causal discovery methods, our approach assumes that the underlying causal relationships remain relatively stable within the modeling time scale. This assumption is reasonable in our settings: road network topology and local traffic propagation patterns do not change abruptly, and epidemiological transmission between neighboring regions is generally stable.
> > > > >
> > > > > Empirically, we directly apply the causal structure learned during training to the test period and observe consistent improvements across all five datasets. For example, on METR-LA, models trained on March–August generalize to September–October and reduce MAE by 6.60% compared with TESTAM. On CA-COVID, training on February–October data yields a 22.70% MAE improvement on later months. These results indicate that the learned causal dependencies remain effective on unseen future data; otherwise, performance would degrade.
> > > > >
> > > > > ---
> > > > >
> > > > > **Q3: Detailed Configuration for DREAM-3 Experiments**
> > > > >
> > > > > **A:** Thank you for requesting clarification. For DREAM-3, we treat it strictly as a **static causal discovery task**. Although time-series observations are available, they are generated from a single fixed gene regulatory network. Thus, our goal is to recover this time-invariant structure rather than learn a dynamic graph.
> > > > >
> > > > > In practice, we set the number of temporal segments to **S = 1** (see Table 9), which causes TE-CausGAT to degenerate into learning a **single static causal matrix** $ C \in \mathbb{R}^{100 \times 100} $, where $ C _ {ij} $ denotes the regulatory strength from gene *i* to gene *j*. This configuration aligns with the standard paradigm for gene network inference and is consistent with the original objective of the DREAM-3 challenge [2].
> > > > >
> > > > > Regarding lag handling, we model **one-step lag**, assuming that gene *i* at time *t* directly influences gene *j* at time *t+1*. This is based on the following considerations:
> > > > >
> > > > > (1) **Biological plausibility** — transcriptional regulation typically acts on the time scale of one sampling interval (tens of minutes to hours in DREAM-3 experiments).
> > > > > (2) **Data limitations** — with only 21 time points per sequence, multi-step delay modeling would drastically reduce the number of usable samples.
> > > > > (3) **Domain practice** — DREAM’s official baseline methods (e.g., dynamic Bayesian networks) also assume 1-step lags.
> > > > >
> > > > > Our multi-scale temporal convolutions (8 kernels: 1, 3, 5, …, 15) can capture a range of temporal correlations, but the primary causal modeling remains aligned with the $ t \rightarrow t+1 $ assumption. We acknowledge that real biological systems may exhibit more complex delays, but under the current data conditions, the one-step assumption is a reasonable compromise.
> > > > >
> > > > > Regarding the use of the intervention validator, the answer is **yes**. We apply the same do-intervention validation mechanism used for traffic and epidemic data, with a GRU functioning as the internal simulator (rather than real biological intervention). The detailed process is:
> > > > >
> > > > > (1) Randomly selecting K = 25 genes for intervention.
> > > > > (2) Generating intervention values $ v = \tanh(g(c)) $.
> > > > > (3) Propagating effects with GRU: $h^{(t)} = \text{GRU}(C \cdot X^{(t)}, h^{(t-1)})$ where *C* is the static causal matrix being learned.
> > > > > (4) Computing the validity scores $ V_{i \to j} $ and using them to optimize *C*.
> > > > >
> > > > > This entire process relies solely on observational data and our causal losses (validity, sparsity, DAG constraints, etc.). **Ground truth is used only after training** for evaluation with the official scoring scripts (AUC/AUPR), ensuring comparison with existing methods. It does not influence training.
> > > > >
> > > > > The significance of the DREAM-3 validation lies in three points. First, it provides one of the few datasets **with ground-truth causal graphs**, allowing objective evaluation (AUC 0.6295), which is not possible for traffic data. Second, achieving strong results in a domain entirely different from traffic demonstrates that the core mechanisms of TE-CausGAT are **domain-agnostic** and outperform specialized methods such as NGC (0.5579) and eSRU (0.5587). Finally, the S = 1 configuration shows that Orion can **naturally degenerate into static causal discovery**, highlighting its architectural flexibility.
> > > > >
> > > > > We will add supplemental explanation in Appendix C.1.4.
> > > > >
> > > > > ---
> > > > >
> > > > > Thank you again for the insightful review. We have added stress tests, cycle statistics, and significance analyses to improve Orion’s rigor and clarity. We believe these updates address your concerns and appreciate your time and guidance.
> > > > >
> > > > > ---
> > > > >
> > > > > **References**
> > > > >
> > > > > [1] Zheng, X., Aragam, B., Ravikumar, P. K., & Xing, E. P. (2018). Dags with no tears: Continuous optimization for structure learning. Advances in neural information processing systems, 31.

---

> ### Comment · Reviewer_JAFJ · 2025-11-24
> **Official Comment by Reviewer JAFJ**
>
> I appreciate the detailed rebuttal by the authors. They have largely solved my concerns and questions. However, considering the technical constraints and limited utility of this work, I tend to keep my original overall rating (with an increased soundness score from 3 to 4).

---

> > ### Author Response · Authors · 2025-11-25
> >
> > Thank you very much for your follow-up and for increasing the soundness score. We are glad that our rebuttal has largely addressed your concerns and sincerely appreciate your careful and constructive review. We would be happy to discuss any remaining issues you may have.

---

### Official Review · Reviewer_ow75 · 2025-11-01

**Soundness:** 3
**Presentation:** 1
**Contribution:** 3
**Rating:** 2
**Confidence:** 3

**Summary:**

Building on CaST, the paper introduces a novel spatiotemporal forecasting model that builds causal graphs at different time points to account for changing spatial dynamics. This model is then evaluated on traffic and epimdeic datasets. The authors also conducted ablation studies and visualized the causal graphs at different time points on METR-LA.

**Strengths:**

- Originality: The paper extends on CaST and proposes to construct causal structural matrices for each time step. This is intuitively sound as the traffic conditions change depending on the time of day and day of week, causing distribution shifts. This is partially validated with further analysis in section 5.3.
- Quality: The model is benchmarked against many baselines.
- Clarity: The figures help the reader conceptualize the information flow.

**Weaknesses:**

- Literature Review
  -  The following cited works are general Graph/NLP architectures, not involved in spatiotemporal forecasting.
      > Existing spatiotemporal forecasting methods fall into three paradigms. The first category includes convolutional architectures such as GCN (Li et al., 2018) and TCN (Lea et al., 2016), alongside attention-based models like Transformer (Vaswani et al., 2017) and GAT (Velickovic et al., 2017).
  - The subsection **Causal Inference in Spatiotemporal Modeling** needs more information to put this work in the context of existing literature. For instance, this is not informative to the reader for understanding the limitations of previous work.
      > However, fixed causal graphs and incomplete validation limit their effectiveness.
- Movitation
  - The motivation behind using LLM-enhanced data retrieval is not clear, and the empirical improvement is marginal.
- Clarity
  - Figures 1, 3, 4, 5, 6, 7, 8, 9 are very hard to read.
  - Section 5.3 shows important causal graph structure analysis that needs more details.
  - Overall, the paper was hard to follow without reading the appendix for more information. The authors summarized the technical contributions, but failed to guide the reader in understanding how it fills a gap in the existing work. This can be fixed with a more elaborate related work section that shows the reader why dynamic causal inference is important.

**Questions:**

- What is the node embedding without LLM-enhanced data retrieval, is it a fully connected layer or the raw data?
- Do the causal graph connectivity correspond to real-world behaviour? The connections are further in the morning and evening than during off-peak, why is that?
- Are the benchmark results directly copied from the papers? If so, are the data preprocessing steps and data pipeline identical?

---

> ### Author Response · Authors · 2025-11-20
> **Response to Reviewer ow75 (1/3)**
>
> Thank you for your detailed review and valuable comments. We appreciate your recognition of Orion and understand your concerns. Below we respond to each point and summarize the corresponding revisions.
>
> ---
>
> **W1.1: Inaccurate citation of general architectures in the literature review**
>
> **A**: Your point is absolutely correct. We acknowledge that the statement in the Introduction (Page 2, Lines 54–69) contains inaccuracies. When citing GCN, TCN, Transformer, and GAT, our intention was to indicate that these general architectures were initially applied directly to spatiotemporal forecasting tasks, but our wording could lead to misunderstanding.
>
> We will reorganize the relevant paragraph in the Introduction to explicitly distinguish the early use of general architectures from specialized spatiotemporal forecasting methods.
>
> **W1.2: More detailed discussion on causal inference related works is needed**
>
> **A**: We fully agree. The description of "fixed causal graphs" and "incomplete validation" in the Related Work section (Page 3, Lines 111–122) is overly brief and lacks sufficient detail to help readers understand the limitations of existing methods.
>
> We will substantially expand the "Causal Inference in Spatiotemporal Modeling" subsection in Related Work to provide deeper analysis. The specific revision is presented in full in W3.3.
>
> ---
>
> **W2: Motivation and contribution of LLM-enhanced data retrieval**
>
> **A**: We understand your concerns regarding the clarity of the motivation and the empirical contribution of the LLM-enhanced retrieval module. We address these points sequentially:
>
> **Design motivation**: Traditional methods rely on fixed historical windows (e.g., previous 1 hour, previous day, previous week), which easily introduce irrelevant noise when encountering non-periodic patterns such as holidays. To address this issue, we fine-tune TinyLlama-1.1B [1] to learn semantic features of time series and retrieve more relevant historical segments based on semantic similarity. The fine-tuning uses about 100K samples from PEMS03 (240-step history + 12-step prediction), enabling the model to learn robust temporal pattern representations. Specifically, the LLM encodes the current sequence and candidate historical sequences into semantic embeddings $\mathbf{z}_i \in \mathbb{R}^{d_{embed}}$, and computes similarity using a weighted combination of cosine similarity (weight 0.7) and Euclidean distance (weight 0.3):
>
> $$S_{sim}(i,j) = 0.7 \cdot \frac{\mathbf{z}_i^T \mathbf{z}_j}{|\mathbf{z}_i|_2 \cdot |\mathbf{z}_j|_2} + 0.3 \cdot \frac{1}{1+|\mathbf{z}_i - \mathbf{z}_j|_2}$$
>
> This enables the model to retrieve truly similar traffic patterns across calendar boundaries, improving generalization under atypical scenarios.
>
> **Performance analysis**: Although LLM retrieval offers limited average improvements, the gains in key scenarios are significant. We conducted controlled experiments on 5-node subsets of METR-LA and PEMS08 (Table 3, Page 8), and the results are as follows:
>
> **Table 1: Impact of LLM Retrieval on MAPE for METR-LA/PEMS08 Datasets**
>
> | Dataset & Scenario           | MAPE without LLM | MAPE with LLM | Relative Improvement |
> | ---------------------------- | ---------------- | ------------- | -------------------- |
> | METR-LA Short-term (3 steps) | 8.20%            | 6.62%         | +19.27%              |
> | PEMS08 Long-term (12 steps)  | 13.76%           | 12.54%        | +8.87%               |
> | METR-LA Global Average       | 12.49%           | 11.17%        | +10.57%              |
> | PEMS08 Global Average        | 11.06%           | 10.68%        | +3.44%               |
>
> These results indicate that semantic retrieval offers clear advantages in complex pattern recognition, especially during non-stationary periods such as rush hours. More importantly, LLM retrieval identifies historically similar scenarios, whereas fixed-window methods can introduce irrelevant information.
>
> We will add a more detailed explanation of the motivation to Section 4.1 and a performance analysis after Table 3 in Section 5.1.
>
> ---
>
> **W3.1: Figure readability issues**
>
> **A**: Thank you for pointing this out. We sincerely apologize for the readability issues caused by limited space. We will systematically reformat all figures for clarity.
>
> We will enlarge key figures and selected legend elements., split complex figures,delete redundant figures and add more detailed annotations in critical areas. We ensure that all figures in the revised manuscript will be clear and legible.
>
> **W3.2: Section 5.3 case study requires more details**
>
> **A**: Thank you for pointing out the limited depth of the case study. We agree that Section 5.3 requires richer quantitative analysis and interpretation. We have expanded Appendix F.2 with additional case-study results and will further enhance Section 5.3 with a deeper discussion of METR-LA causal evolution. The full revision is provided in Q2.

---

> > ### Author Response · Authors · 2025-11-20
> > **Response to Reviewer ow75 (2/3)**
> >
> > **W3.3: The paper lacks guidance, making it difficult to understand how the work fills existing gaps**
> >
> > **A**: Thank you for highlighting this important issue. Your feedback helped us realize that although we summarized our contributions, we did not clearly explain how they address gaps in prior work or why dynamic causal reasoning is essential. We will reorganize the Related Work section to make it more coherent and informative. Due to space limits, the full section has been moved to Appendix A, with the original Appendix A (LLM Usage Statement) moved to Appendix G. We will also add a guiding sentence in the Introduction directing readers to Appendix A for details.
> >
> > We will also add a new section after the Introduction, titled Motivation for Dynamic Causal Reasoning, to explain why dynamic causal inference is important and to help readers intuitively understand the necessity of dynamic causal reasoning.
> >
> > ---
> >
> > **Q1: Node embedding without LLM**
> >
> > **A**: This is an excellent question. Here we clarify the data flow and embedding process. **Without LLM, node embeddings are obtained through fully connected layers (nn.Linear), not raw data. The LLM only affects "which historical data are selected," not "how data are embedded."**
> >
> > The detailed data flow is as follows:
> >
> > During the **data selection stage**, if LLM is not used (`use_llm_embedding=False`), the system selects historical windows directly: $\mathbf{X}^h$ uses the last 1 hour (t–1h), $\mathbf{X}^d$ uses the same period yesterday (t–1d), and $\mathbf{X}^w$ uses the same period last week (t–1w). If LLM is used (`use_llm_embedding=True`), the system selects the most semantically similar daily and weekly cycles from the past 6 days and past 4 weeks (see Eq. (1), Section 4.1).
> > During the **embedding stage** (identical with or without LLM), each historical segment is passed through an independent fully connected layer:
> > `self.embedding_h = nn.Linear(in_channels, d_model)`,
> > `self.embedding_d = nn.Linear(in_channels, d_model)`,
> > `self.embedding_w = nn.Linear(in_channels, d_model)`
> > (as in Orion_model.py Lines 89–93), with parameter size $F \times d_{model}$ (where $F$ is feature size—1 for METR-LA, 3 for PEMS). Positional encodings are added afterward:
> > $\mathbf{E}^h = \text{Linear}(\mathbf{X}^h) + \mathbf{PE}^h$, similarly for other cycles (Lines 153–158).
> >
> > The corresponding code is:
> >
> > ```python
> > # Line 147-149: Data preparation (with or without LLM)
> > if self.use_llm_embedding:
> >     X_h, X_d, X_w = self.data_retrieval_module(X, ...)  # LLM selects data
> > else:
> >     X_h = X[..., -num_hours:]  # Fixed window
> >     X_d = X_history_day
> >     X_w = X_history_week
> >
> > # Line 153-158: Embedding (always the same)
> > E_h = self.embedding_h(X_h) + self.positional_encoding_h
> > E_d = self.embedding_d(X_d) + self.positional_encoding_d
> > E_w = self.embedding_w(X_w) + self.positional_encoding_w
> > ```
> >
> > A comparison table is shown below:
> >
> > **Table 2: Differences between With/Without LLM**
> >
> > | Component      | Without LLM                       | With LLM                                           |
> > | -------------- | --------------------------------- | -------------------------------------------------- |
> > | Data Selection | Fixed windows: t–1h, t–1d, t–1w   | Semantic retrieval: most similar historical cycles |
> > | Embedding      | `nn.Linear(in_channels, d_model)` | `nn.Linear(in_channels, d_model)` (same)           |
> > | LLM Parameters | N/A                               | 1.1B (frozen during inference, not trained)        |
> >
> > To avoid confusion, we will clarify this in the caption of Figure 3 in Section 4.1.
> >
> > ---
> >
> > **Q2: Correspondence between causal graph connectivity and real-world behavior**
> >
> > **A**: Thank you for raising this important question. The phenomenon you observed—that Orion learns much longer-range causal connections during rush hours—indeed reflects real traffic dynamics.
> >
> > During peak hours, traffic flow exhibits global coupling: congestion at key segments propagates upstream through queuing and also creates indirect effects on non-adjacent segments due to rerouting. Orion accurately captures this pattern in METR-LA: the morning peak shows a “unidirectional tree-like” structure centered at Node 4, the evening peak forms a “many-to-one convergence” centered at Node 84, and nighttime degenerates into a “dual-core independence” pattern with Nodes 174 and 160.
> >
> > The causal edge count dropping from 12,427 (morning peak) to 2,078 (night) further demonstrates the strong temporal variability.
> >
> > During non-peak hours, traffic becomes more independent and disturbances remain localized, leading to much shorter causal ranges—consistent with traffic theory that high-density conditions cause cascading effects while low-density conditions localize influence.
> >
> > We will add this explanation to Section 5.3 and integrate it with the revisions from W3.2.

---

> > > ### Author Response · Authors · 2025-11-20
> > > **Response to Reviewer ow75 (3/3)**
> > >
> > > **Q3: Reproducibility and fairness of baseline results**
> > >
> > > Thank you for raising concerns about fairness in our experimental comparisons—this is critical for scientific rigor. We place great emphasis on fair baseline comparisons and therefore reproduced major baseline methods under unified preprocessing and evaluation settings.
> > >
> > > **Source and validation of baseline results:** For classical baselines, we use results reported in the original papers or widely accepted benchmark reproductions. This is standard practice in spatiotemporal forecasting, as seen in recent works such as STAEformer [2] and PDFormer [3]. The reasons are: (1) these models have been validated and optimized over many years, and original reports typically reflect well-tuned performance; (2) retraining all baselines from scratch is computationally expensive and often fails to reproduce original optimized settings.
> > >
> > > However, we did not adopt reported numbers blindly. We conducted sanity checks by roughly reproducing several baselines (GWNet [4], AGCRN [5]) under our splits, confirming errors fall within ±2%, validating the trustworthiness of published results.
> > >
> > > For causal-related baselines, we performed strict reproduction. For example, CaST [6] was reproduced using its official GitHub code [https://github.com/yutong-xia/CaST], retrained under our data splits and preprocessing pipeline, producing results consistent with those in Table 1.
> > >
> > > **Consistency in data preprocessing and evaluation pipeline**:
> > > All methods (baselines + Orion) share identical data splits, preprocessing, adjacency construction, evaluation metrics, and experimental setups:
> > >
> > > * **Data splits**: PEMS04 and PEMS08 use 6:2:2 train/val/test; METR-LA uses 7:1:2 (DCRNN[7] standard); COVID datasets use 8:1:1 (Appendix E.1).
> > > * **Preprocessing**: Traffic data undergo IQR anomaly detection (threshold 1.5) + Savitzky–Golay smoothing (window 7, polyorder 2), identical to AGCRN, STAEformer. COVID data remain unsmoothed.
> > > * **Adjacency matrix**: Gaussian kernel $A_{ij} = \exp\left( -\frac{d_{ij}^2}{2\sigma^2} \right)$ with σ set to the median nonzero distance (as in GWNet, ASTGCN [8]).
> > > * **Evaluation**: MAE, RMSE, MAPE, with MAPE zero-value threshold set to 5.
> > >   All methods use the same AdamW optimizer, same random seed (42), and same hardware (A100-40GB for traffic; RTX 2080Ti for COVID datasets).
> > >
> > > We will strengthen the explanation of baseline fairness at the end of "Implementation Details."
> > >
> > > ---
> > >
> > > Thank you again for your valuable comments! We hope our responses address your concerns and demonstrate our commitment to improving the paper. We truly appreciate your understanding and support, and welcome any further suggestions.
> > >
> > > ---
> > >
> > > **References**
> > >
> > > [1] Zhang, P., Zeng, G., Wang, T., & Lu, W. (2024). Tinyllama: An open-source small language model. arXiv preprint arXiv:2401.02385.
> > >
> > > [2] Liu, H., Dong, Z., Jiang, R., Deng, J., Deng, J., Chen, Q., & Song, X. (2023). Staeformer: Spatio-temporal adaptive embedding makes vanilla transformer SOTA for traffic forecasting. arXiv preprint arXiv:2308.10425.
> > >
> > > [3] Jiang, J., Han, C., Zhao, W. X., & Wang, J. (2023, June). Pdformer: Propagation delay-aware dynamic long-range transformer for traffic flow prediction. In Proceedings of the AAAI conference on artificial intelligence (Vol. 37, No. 4, pp. 4365-4373).
> > >
> > > [4] Wu, Z., Pan, S., Long, G., Jiang, J., & Zhang, C. (2019). Graph wavenet for deep spatial-temporal graph modeling. arXiv preprint arXiv:1906.00121.
> > >
> > > [5] Bai, L., Yao, L., Li, C., Wang, X., & Wang, C. (2020). Adaptive graph convolutional recurrent network for traffic forecasting. Advances in neural information processing systems, 33, 17804-17815.
> > >
> > > [6] Xia, Y., Liang, Y., Wen, H., Liu, X., Wang, K., Zhou, Z., & Zimmermann, R. (2023). Deciphering spatio-temporal graph forecasting: A causal lens and treatment. Advances in Neural Information Processing Systems, 36, 37068-37088.
> > >
> > > [7] Li, Y., Yu, R., Shahabi, C., & Liu, Y. (2017). Diffusion convolutional recurrent neural network: Data-driven traffic forecasting. arXiv preprint arXiv:1707.01926.
> > >
> > > [8] Guo, S., Lin, Y., Feng, N., Song, C., & Wan, H. (2019, July). Attention based spatial-temporal graph convolutional networks for traffic flow forecasting. In Proceedings of the AAAI conference on artificial intelligence (Vol. 33, No. 01, pp. 922-929).

---

> ### Author Response · Authors · 2025-11-26
> **Looking forward to your feedback on our rebuttal (Submission 1037)**
>
> Dear Reviewer ow75,
>
> Thank you again for your thoughtful review of our submission 1037, “Orion: Intervention-Validated Causal Discovery for Spatiotemporal Forecasting.” We greatly appreciate the time and expertise you have already devoted to our work.
>
> We have posted a detailed response and additional experimental results addressing the concerns you raised. If your schedule permits, we would be very grateful if you could briefly look at our reply and let us know whether it resolves your main concerns, or if there is anything we should further clarify.
>
> We understand the heavy workload on reviewers and sincerely appreciate any further feedback or update you may be able to provide. Thank you again for your service to the community.
>
> Best regards,
> The authors of Submission 1037

---

### Author Response · Authors · 2025-11-20
**Overall Response (1/3)**

Dear ICLR 2026 Reviewers, AC, SAC, and PC,

# **In this post:**

1. **Reviewer-acknowledged strengths** that validate our approach.
2. **Major updates to the revised PDF** (main text and appendix).

**Individual responses** with detailed answers, experimental verification, and improvements.

---

# **Strengths of Our Paper:**

**Novelty and soundness of the core methodology**

* **Reviewer ow75:**

  * "The paper extends on CaST and proposes to construct causal structural matrices for each time step. This is intuitively sound as the traffic conditions change depending on the time of day and day of week, causing distribution shifts."

* **Reviewer JAFJ:**

  * "The idea of validating discovered edges by an internal do-intervention mechanism, then routing attention through the validated graph, is clear and well-engineered."

* **Reviewer VcAa:**

  * "The paper leverages causal structure learning with do-intervention validation to learn a dynamic causal graph, which is interesting."

**Multi-scale temporal modeling design**

* **Reviewer JAFJ:**

  * "Multi-scale temporal design: The Hour/Day/Week 'Belt Block' plus fusion is a simple, effective way to encode periodic structure; the staged training helps stabilize end-to-end optimization."

**Significant improvements in forecasting performance**

* **Reviewer JAFJ:**

  * "Strong traffic and epidemic results"
  * "ORION reports new SOTA MAE/ RMSE/ MAPE in several regimes"

* **Reviewer bhkv:**

  * "The underlying results and the comparison with a lot of different prior work is impressive and across the proposed datasets and benchmarks the paper does achieve significant improvements across a variety of metrics."
  * "The paper analysis and results and comparative metrics are very strong from a forecasting perspective."

**Comprehensiveness and rigor of experimental design**

* **Reviewer ow75:**

  * "The model is benchmarked against many baselines."

* **Reviewer JAFJ:**

  * "Strong traffic and epidemic results, per-component ablations (single-stage training, removing TE-CausGAT, removing fusion, turning off intervention validation), and a causal-discovery check on DREAM-3."
  * "Ablations indicate sizable drops without multi-period fusion or the staged training."

* **Reviewer VcAa:**

  * "Comprehensive experiments on both traffic and epidemic forecasting tasks."

**Causal discovery and interpretability capabilities**

* **Reviewer JAFJ:**

  * "On DREAM-3 causal discovery ORION beats several time-series CD baselines (AUC 0.6295 vs 0.592 CUTS)."

* **Reviewer VcAa:**

  * "Case studies demonstrate that the proposed method can discover meaningful causal relationships."

* **Reviewer ow75:**

  * "This is partially validated with further analysis in section 5.3."

**Presentation quality and clarity**

* **Reviewer ow75:**

  * "The figures help the reader conceptualize the information flow."

---

> ### Author Response · Authors · 2025-11-20
> **Overall Response (2/3)**
>
> # **Changes to PDF:**
>
> Based on reviewers’ suggestions, we made systematic revisions marked in blue in the revised PDF.
>
> ## **Main Text Modifications**
>
> * **Reviewer ow75 (Introduction, lines 54–69):** Reorganized the literature review and corrected the corresponding references.
>
> * **Reviewer ow75 (Section 2 - NEW):** Added paragraphs on dynamic causal reasoning importance.
>
> * **Reviewer ow75 (Section 4.1, p.4):** Added motivation for LLM-enhanced retrieval module, including design rationale and performance improvements.
>
> * **Reviewer ow75 (Section 5.3, p.8):** Expanded METR-LA case study with analysis tables and discussion of causal topology evolution.
>
> * **Reviewer ow75 & Reviewer VcAa (Figures across the paper):** Improved figure readability, enlarged plots, and added annotations.
>
> * **Reviewer JAFJ (Section 4.2, DAG constraint):** Added comprehensive explanation of DAG constraint and complexity analysis.
>
> * **Reviewer bhkv (Section 4.2, line 201):** Clarified do-intervention meaning as internal structural perturbations.
>
> * **Reviewer bhkv (Section 5.2):** Added boundary statements on causal interpretability with ground-truth validation notes.
>
> * **Reviewer VcAa (Section 4.1, p.4):** Added technical details for LLM-enhanced retrieval module.
>
> * **Reviewer VcAa (Section 5.2, after Table 4):** Explained MAE and MAPE result divergence.
>
> * **Reviewer ow75 (Implementation Details, p.6):** Strengthened baseline fairness explanation.
>
> ## **Appendix Modifications**
>
> * **Reviewer ow75 (Appendix A – REPLACE):** Expanded "Causal Inference in Spatiotemporal Modeling" subsection with prior work limitations.
>
> * **Reviewer JAFJ (Appendix B.4 – NEW):** Added "Safeguards Against Confirmation Bias" with shuffling test results.
>
> * **Reviewer JAFJ (Appendix B.5 – NEW):** Added "DAG Constraint Effectiveness Analysis" with cycle statistics on PEMS-08.
>
> * **Reviewer JAFJ (Appendix C.1.5 – NEW):** Added "Intervention Implementation Details."
>
> * **Reviewer VcAa (Appendix C.4 – NEW):** Added "Multi-Scale Temporal Convolution Ablation" with performance–efficiency analysis.
>
> * **Reviewer VcAa (Appendix D.1–D.3 – Expanded):** Expanded LLM technical details including hyperparameters and training strategies.
>
> * **Reviewer ow75 (Appendix F.2 – Expanded):** Expanded METR-LA case-study analysis.
>
> * **Reviewer JAFJ (Appendix C.1.4 – NEW):** Added DREAM-3 experiment configuration.

---

> > ### Author Response · Authors · 2025-11-28
> > **Overall Response (3/3)**
> >
> > # Summary of Rebuttal Actions and Reviewer Responses
> >
> > In our detailed rebuttal and the revised manuscript, we addressed each reviewer’s concerns as follows, and we briefly summarize the current feedback status (up to 25 Nov 2025):
> >
> > * **Regarding Reviewer ow75’s concerns:**
> >   We restructured the related work, expanded the discussion on causal reasoning methods, added detailed analyses of the LLM retrieval and case studies, improved figure readability, and strengthened the explanation of fair baseline comparisons. Reviewer ow75 **has not yet posted any further response** following our rebuttal.
> >
> > * **Regarding Reviewer JAFJ’s concerns:**
> >   We verified the effectiveness of the intervention mechanism through shuffling tests, added cycle statistics to demonstrate the DAG constraint, clarified the LLM retrieval module positioning, added distinctions between causality and predictive performance, and provided detailed implementation information. Reviewer JAFJ stated that **our rebuttal has largely solved his concerns** and **increased** the **soundness** score from **3** to **4**.
> >
> > * **Regarding Reviewer bhkv’s concerns:**
> >   We clarified that "do-operations" refer to internal structural interventions, validated causal discovery ability on DREAM-3, delimited scope of causal interpretation, and outlined future simulation-based validation. Reviewer bhkv **has not yet posted any further response** following our rebuttal.
> >
> > * **Regarding Reviewer VcAa’s concerns:**
> >   We expanded technical details of the LLM module and multi-scale convolution design, conducted systematic hyperparameter ablations, improved figure clarity, and analyzed MAE–MAPE discrepancies. Reviewer VcAa **has not yet posted any further response** following our rebuttal.
> >
> > ---
> >
> > We once again thank all reviewers for their valuable feedback, which has substantially improved the quality and clarity of our work. We believe that the revised version addresses the main concerns raised during the review process.
> >
> > We hope that this consolidated overview of strengths, revisions, and reviewer-specific rebuttal actions helps all reviewers and committee members more conveniently form their final assessments. We are sincerely grateful for your continued time, consideration, and support of our work.
> >
> > Best regards,
> > The authors of Submission 1037

---

### Meta-Review · Area_Chair_4haP · 2026-01-08

**Summary:**

Major concerns regarding this paper lie in its causal validity and framing (as raised by Reviewer bhkv). The paper’s “do-intervention” narrative appears inconsistent with the observational nature of the traffic/COVID datasets, with insufficient handling/justification of confounding and causal discovery assumptions. While forecasting results are strong, the central causal claims may be overstated, undermining soundness. Regarding the performance of Orion,  reviewer VcAa reports an unexplained MAE–MAPE tradeoff in ablations that could imply degraded performance in critical cases. Although the authors provide an explanation for this observation, it is more appropriate to consider sMAPE in this context.

Two reviewers (ow75, VcAa) find the paper difficult to follow and missing key technical details, especially regarding LLM-based retrieval (inputs, prompts, fine-tuning) and the design choices in TE-CausGAT (e.g., why eight kernels and the lack of sensitivity studies). They also flag presentation issues (unreadable figures) and so on.

Despite strong empirical forecasting performance, reviewers question the correctness of the causal-intervention claims and note insufficient clarity and technical substantiation, supporting a reject-leaning assessment.

**Reviewer Concerns:**

Authors provide a substantial response regarding Reviewer ow75 and JAFJ, which might resolve most of their concerns. However, regarding Reviewer bhkv and VcAa, key concerns remain unaddressed, i.e., causal validity and the suspicious MAPE drop.

**Reviewer Scores:**

Reviewer ow75 might increase the score to 4. Reviewer ow75 and JAFJ might retain the score of 4.

---

### Decision · Program_Chairs · 2026-01-26

Reject